# Silver Nanowire Synthesis and Strategies for Fabricating Transparent Conducting Electrodes

**DOI:** 10.3390/nano11030693

**Published:** 2021-03-10

**Authors:** Amit Kumar, Muhammad Omar Shaikh, Cheng-Hsin Chuang

**Affiliations:** 1Institute of Medical Science and Technology, National Sun Yat-sen University, Kaohsiung 80424, Taiwan; amitkumar82829@gmail.com; 2Sustainability Science and Engineering Program, Tunghai University, Taichung 407, Taiwan

**Keywords:** nanowire, transparent conducting electrodes, optoelectronics

## Abstract

One-dimensional metal nanowires, with novel functionalities like electrical conductivity, optical transparency and high mechanical stiffness, have attracted widespread interest for use in applications such as transparent electrodes in optoelectronic devices and active components in nanoelectronics and nanophotonics. In particular, silver nanowires (AgNWs) have been widely researched owing to the superlative thermal and electrical conductivity of bulk silver. Herein, we present a detailed review of the synthesis of AgNWs and their utilization in fabricating improved transparent conducting electrodes (TCE). We discuss a range of AgNW synthesis protocols, including template assisted and wet chemical techniques, and their ability to control the morphology of the synthesized nanowires. Furthermore, the use of scalable and cost-effective solution deposition methods to fabricate AgNW based TCE, along with the numerous treatments used for enhancing their optoelectronic properties, are also discussed.

## 1. Introduction

One-dimensional (1D) metal nanomaterials have attracted widespread research interest due to their unique electrical, magnetic, optical, thermal and catalytic properties. Consequently, such nanomaterials have become promising candidates for a wide range of applications in optoelectronics, nanoelectronics, nanophotonics and micromechanics because of their exceptional density of states and high aspect ratio [1,2,3,4,5]. In particular, silver (Ag) has the highest electrical and thermal conductivities of all metals, and its 1D form, referred to as Ag nanowires (AgNWs), has huge potential for practical applicability in a range of technologies. Over the years, numerous AgNW synthesis protocols have been proposed, including the use of soft and hard templates and a range of wet-chemical techniques like hydrothermal, solvothermal and polyol-based synthesis. This has resulted in wide range of achievable dimensions (diameter 2–300 nm and length 1–500 µm). A relative comparison on the bases of simplicity, control, yield and cost demonstrated that the polyol process is the most promising method for AgNW synthesis. Research in the area of AgNWs has been extensive, with over 100,000 publications covering a vast range of applications including smart sensors [6,7,8,9,10], catalysis [11], energy harvesting devices [12], optoelectronic displays [13] and wearable electronics [14,15,16], among others. Furthermore, theoretical predictions and experimental demonstrations have highlighted the fact that AgNWs can significantly improve the properties of transparent conducting electrodes (TCE), which are an essential component of optoelectronic devices. Traditionally the material of choice for fabricating TCE has been indium tin oxide (ITO) due to its low sheet resistance, high optical transmittance, low surface roughness and ability to pattern electrodes using photolithography. However, ITO has various drawbacks, including its high fabrication cost and brittle nature, which hinders its applicability for use in next-generation flexible optoelectronic devices like smart windows [17,18], electronic skins [19], smart clothes [20], wearable heaters [21,22], foldable displays [23,24,25,26], compliant solar cells [27] and flexible display panels [28]. These devices require TCE that fulfill certain criteria, including solution-based low cost processing capability, excellent optoelectronic properties, mechanical compliance and chemical stability. A variety of alternatives to ITO, such as carbon nanotubes (CNTs), graphene oxide (GO), aluminum zinc oxide (AZO), metal nanowires, nanofibers and conductive polymers have been reported for TCE fabrication. In particular, AgNW based TCE have emerged as a perfect contender for modern optoelectronic and wearable devices due their low-cost and scalable solution based processing, high conductivity and transparency, smooth surface and mechanical flexibility [16,29,30,31,32,33,34,35,36,37]. AgNWs can be deposited on various substrates and form a network with openings between the nanowires that allow light transmittance while conductive pathways are formed at the nanowire junctions.

Herein, we have compiled and presented a detailed review of AgNW synthesis and strategies for AgNW based TCE fabrication. A schematically illustrated outline of the review is presented in Figure 1. To the best of our knowledge, this is the first review that discusses all solution based TCE fabrication methods in detail and provides a comprehensive outlook of the various treatment techniques utilized to improve TCE performance. Briefly, this review is structured in the following manner: Section 1 introduces AgNWs and their utilization for TCE fabrication. Section 2 discusses the numerous AgNW synthesis methods, including template assisted and wet chemical techniques. Section 3 discusses the various solution-based methods used for TCE fabrication. In addition, it also provides a detailed analysis of the various pre- and post-treatment methods used to improve the optoelectronic properties of the fabricated TCE. Finally, outlooks and challenges for the practical applicability of AgNWs are presented.

## 2. AgNW Synthesis

Over the years, several protocols for AgNW synthesis have been reported. The use of templates, such as hard and soft templates, were widely investigated in the early years of AgNW synthesis. While predefined dimensions of the template assist in uniform nanowire synthesis, purification methods are quite complex and result in low yields. To overcome these issues, various wet chemical methods such as hydrothermal, solvothermal and polyol were later introduced. Among these, the polyol method has gathered much attention, since it is facile, scalable and provides good control over nanowire morphology. In this section, first, we will discuss the template methods, followed by the wet chemical methods. A Table 1 summarizing the various synthesis protocols is also added in the end of this section to provide a concise comparison and improve readability.

### 2.1. Template Methods

Researchers have utilized a range of templates that enable the growth of 0-D nanoparticles into 1-D nanowires. These templates are generally used to provide a specific size and structure to the nanowire [38]. These templates can be broadly categorized as either hard or soft. Hard templates result in greater control over the shape, size and overall morphology of the nanowire because of their predesigned structure that shapes the Ag atoms into nanowires. Although simple and effective, there can be significant loss of generated nanowires during the purification step, leading to lower yields. On the other hand, soft template methods utilize chemicals which are dispersible/dissolvable in solvents. Consequently, the synthesized nanowires can be easily purified from the solvent phase, thus resulting in improved efficiency and scalability.

#### 2.1.1. Hard Template Method

The hard-template method utilizes a material of nanoscale porosity, which primarily serves as a template for the targeted material to grow in a well-designed manner. The template materials can be nanoporous polymers, carbon nanotubes (CNT), nonporous silica [39] and zeolites, which direct the germination and synthesis of AgNWs [40,41,42,43]. The idea of using host matrices of CNT was first introduced by Ajayan [40]. They first annealed the CNT in the presence of liquid lead, which opened the capped tube ends, and then filled these tubes with molten silver through capillary action. Following the same path, Ugarte et al. [41] described a new way to fill the CNT with silver to produce AgNWs having a length of 120 nm. Silver nitrate (AgNO_3_) salt was thermolyzed by an electron irradiation beam to obtain pure molten silver, which was injected into the CNT under high pressure to form nanowires. Han and coworkers demonstrated a simple procedure for the formation of nanowires using mesoporous silica SBA-15 as the template [42]. The growth direction and diameter of AgNW was easy to control in this method due to the highly ordered pore size distribution of SBA-15. Takai et al. also synthesized AgNWs with the help of mesoporous silica (SBA-15) powders and dimethyl-amine borane (DMAB) as the reducing agent [43]. This enabled the AgNWs to grow in the micro pores of the SBA-15 by capillary force under low pressure as schematically shown in Figure 2a. The SBA-15 templating method can be used to produce AgNWs with aspect ratios greater than 500 (L = 4 µm, D = 7 nm) as shown in the TEM images in Figure 2f,g [44].

DNA has also been utilized a template due to its useful mechanical [45,46] and molecular recognition properties [47]. The mechanical properties allow it to make the required shape, alignment and junctions, while the recognition property improves the selectivity of Ag ions to form supramolecular structures [48]. A *λ*-DNA based template was introduced between two gold electrodes separated by a few micrometers; the resulting *λ*-DNA bridge could selectively choose the Ag ions by ion exchange process to form a wire-like structure [49]. A further addition of acidic hydroquinone solution reduced the Ag to form a more conductive aggregate. A schematic of the procedure is shown in Figure 2c, and the fabricated AgNWs with a length of 12–16 µm were characterized by AFM and fluorescence imaging, as shown in Figure 2d. Berti and coworkers also utilized a *λ*-DNA-template and photoreduction process to obtain silver composites and films [50]. The synthesized silver ions (Ag^+^) were embedded into the *λ*-DNA to form a complex, which, after photoreduction, was reduced to Ag nanoparticles, as confirmed by the peak observed at 420 nm using UV-VIS spectroscopy. Lu et al. demonstrated that sunlight could also be used for the reduction of silver ions to nanoparticles adsorbed on the surface of DNA [51]. Among all the templates, anodic aluminum oxide (AAO) is the most commonly used for nanowire synthesis [52,53]. In 2020, AAO templates were used to synthesize single crystalline AgNWs, as shown in Figure 2e [54]. The process used molten Ag and the AgNWs were grown in bundles at 970 °C under capillary action. Hydrothermal etching was later used to remove the AAO template using sodium hydroxide (NaOH) solution in a high pressure autoclave, as schematically illustrated in Figure 2b.

In summary, AgNW synthesis can be easily controlled to get the desired shape and size by the hard template method. Furthermore, the synthesized nanowires are already in a composite state, which can aid in practical applicability like device fabrication. However, to enable purification, dissolution of template is required, during which the AgNWs suffer from unfavorable conditions, and their high active surface energy often leads to damage in form of oxidation and etching.

#### 2.1.2. Soft Template Methods

The quantity and quality of the synthesized nanowires via hard template methods is restricted to the porosity and interconnectivity of template. In addition, the harvesting step of nanowires is not viable for practical application at a large scale. Instead, the availability of soft template methods makes processing better and more efficient. Various amphiphilic compounds, such as calix-hydroquinone (CHQ), Supramolecular *β*-cyclodextrin (*β*-CD), cholesteryl pyridine carbamate (CPC) organogel, double-hydrophilic block copolymers (DHBC) and J-aggregate dye, support the generation of self-assembled templates for nanowire synthesis.

Hong and coresearchers used the soft template method to synthesize highly stable and single crystal ultrathin AgNWs with atomic dimensions (0.4 nm diameter) and length in the micrometer range. They used the pores of CHQ, because HQs are known for their reducing properties [55]. Ag ions were reduced to form silver aggregates by electro-/photochemical reaction in solution phase at ambient conditions (temperature and pressure). The synthesized nanowires could be kept in ambient temperature and pressure for one month with no change in the properties. The idea of using polymers as a soft template may also be viable due to the longer structural length of polymer. Hence, a novel approach for nanowire formation in a Poly-vinyl alcohol (PVA) film by electrodeposition was reported [56]. They performed two stage treatment of PVA film with ammonium persulphate [(NH_4_)_2_S_2_O_8_] and pyrrole (C_4_H_4_NH) to enable polymerization. Later, they soaked this film in a AgNO_3_ precursor solution, which acted as an electrolyte, and the nanowires grew in the nanoscale channels of the polymer via electrodeposition. Organogels, which is a 3-D cross-linked matrix composition of liquid organic phase, especially stimuli-responsive gels, have wide variety of effective utilization in targeted drug-delivery systems and biomaterials [57,58,59]. Researchers are interested is using these supramolecular gels as soft templates for AgNW synthesis [60,61]. In 2018, *β*-CD derived soft templates were used for synthesis of AgNWs in the presence of iron chloride (FeCl_3_) [60]. FeCl_3_ primarily helps in formation of longer nanowires by circumventing the oxidation of the AgNW surface. An optimized ratio of FeCl_3_ and AgNO_3_ (1:13.3) with a 0.23 M concentration of soft template yielded nanowires with a length of up to 20 µm and a diameter of about 65 nm, as shown in Figure 3a,b.

Nguyen and Liu introduced a novel method in 2015 to synthesize long AgNWs via self-assembly of CPC [59]. The CPC was used to create highly stable nanotubes (longer than 100 micrometers) and acted as a soft template by providing the skeleton for the growth of AgNW through them in a gel of water and anisole. A schematic for the AgNW synthesis using CPC is depicted in Figure 3c. Although the AgNW synthesis was one-step, an optimized protocol with control over the AgNO_3_ concentration, reduction rate and gel synthesis ratio of water and anisole was extremely crucial for synthesis of long nanowires. Double-hydrophilic block copolymers (DHBC) consist of two hydrophilic polymers, whereby one interacts with the material and the other enhances the solubility [63]. Zhang et al., for the first time, used polymethyl methacrylate (PMMA) and polyethylene oxide (PEO) as DHBC for the synthesis of AgNWs at room temperature [64]. PEO enhanced the solubility of polymer in water solution and remained noninteractive with AgNO_3_, while PMMA helped in Ag reduction to form silver nanoparticles, which slowly grew into nanowires over time. J-aggregates are a special type of dye that have the property of supramolecular self-organization. Interestingly, some researchers use double-walled nanotubular J-aggregates of amphiphilic cyanine dyes as template, which self-assemble in a mixture of water and ethanol to form double walled nanotubes, as shown in Figure 3d [62]. These reversible J-aggregate soft templates provide the structure for nanowire growth in a very small diameter, i.e., less than 7 nm, as shown in the TEM image in Figure 3e,f, with the cyanine dye layer acting as the reducing agent for silver ions in photoinitiated conditions. The purification steps for the nanowires were quite easy due to the reversibility of soft template by surface passivation. These nanotubes provided more control over growth, and provided a better alternative for easy removal of the template after nanowire synthesis.

### 2.2. Chemical Methods

Most of the template methods are not efficient for scalable industrial production of AgNWs. This has encouraged researchers to think beyond existing templates and develop chemical synthesis methods that provide better control over the obtained morphology and avoid the template removal step. Chemical methods include solvents, surfactants, reducing and oxidizing agents to enable nanowires synthesis. The first stage of synthesis is nanocrystal formation in the solution, which later grows to form the nanorod (aspect ratio <10) or nanowire (aspect ratio ~10^1^–10^4^) structure. The process of synthesizing AgNWs and nanorods by Ag seed in controlled manner are quite similar which can be observed in Figure 4h,i. Ag seeds were synthesized by reducing AgNO_3_ in presence of Sodium borohydride (NaBH_4_) and trisodium citrate (Na_3_C_6_H_5_O_7_) where NaBH_4_ controls the particle size and Na_3_C_6_H_5_O_7_ stabilizes it. Later to synthesize nanowires and nanorods, ascorbic acid (AA) was used to reduce AgNO_3_ in a solution embedded with Ag seed, NaOH and the micelle template cetyltrimethylammoniun bromide (CTAB) [65]. The relative amount of NaOH in solution was the deciding factor for the synthesis of nanowires and nanorods. Since the optimized pH for nanorods and nanowires were 11.8 and 4.1 respectively, it suggests that the monoanion of ascorbic acid is the dominating factor in nanowire synthesis while the ascorbate dianion is an important factor in nanorod synthesis.

Sonochemical methods have also been used for the synthesis of AgNWs [66,67]. J.-J. Zhu et al. proposed a novel and seedless method for synthesis of Ag nanowires by a sonoelectrochemical method [68]. The electrochemical cell was composed of two platinum (Pt) sheets and an electrolyte composed of AgNO_3_ and ethylenediamine tetraacetic acid (EDTA) in water. The whole cell was kept in ultrasonic bath under nitrogen (N_2_) atmosphere with a current and voltage controlled electrolysis process for 30–45 min to grow AgNWs at 30 °C temperature. Another wet chemical synthesis method of AgNW interlacing bunches on glass walls by mild chemical reduction in aqueous solutions of PMAA was demonstrated [69]. Single-crystalline AgNWs with diameters of about 30–40 nm were synthesized inside the glass tube after 24 h at ambient temperature in the presence of AA as a reducing agent. Liu et al. introduced the idea of seed and crystals, where they first synthesized the silver bromide (AgBr) crystal seed which later grows into AgNWs with a maximum length of 9 µm [70].

#### 2.2.1. Hydrothermal Methods

Hydrothermal synthesis methods consist of crystallizing the nanomaterial at high temperature and pressure in aqueous solution [71]. Hydrothermal methods for the synthesis of AgNWs are not widely applied these days, but in the first decade 21st century, hydrothermal processing produced the best nanowires. This method does not need any surfactant and copolymers as solvent or reducing agents. It is, therefore, a more favorable green chemistry route for growth of very long nanowires of length up to 500 µm without using any organic solvent or hazardous elements. Primarily, sodium chloride (NaCl) and AgNO_3_ are mixed in water to make silver chloride (AgCl). The Ag ions from AgCl are available during the growth process which lasted for 18 h at 180 °C in an autoclave [72]. Their idea of using glucose as a soft reducing agent is justified as it is also widely used for depositing silver films on glass substrate. The crystallinity and the growth direction of AgNW (perpendicular to the 200 plane) is shown in Figure 4c.

Xu and group demonstrated Ag nanowire synthesis by hydrothermal method at lower temperature (100 °C) in an autoclave with the help of 1,3-bis (cetyldimethylammonium) propane dibromide as the reducing agent over a time span of one day [73]. The problem assisted with the hydrothermal methods was precipitation of AgCl and Ag nanocrystals, which can be seen in Figure 4a,b. In order to avoid the precipitation effect, Bari et al. used Polyvinylpyrrolidone (PVP) as a surfactant and made hydrosol with water and D+ glucose, which enhanced the anisotropic growth property and morphology of the AgNWs [74]. The PVP addition results in a precipitationless solution (Figure 4d); however, without PVP, a precipitate appears, as seen in Figure 4e. The process involves drop wise addition of NaCl ions to promote the growth in a very stabilized and controlled way. They put the hydrosol in a stainless steel autoclave for 22 h at 160 °C. Very-long and thin AgNWs of length in the range of 200 to 500 µm with an average diameter of 45–65 nm were synthesized with a uniform surface as shown in Figure 4g,h.

#### 2.2.2. Polyol Method

• Historical development

Among the available methods, the polyol method is the most widely utilized by researchers. Currently, there are more than 10,000 publications on AgNW synthesis using polyol method in last decade. Fievet et al. introduced the benefits of polyol process for metal nanostructure synthesis [75]. The reason for adopting this method is simplicity, cost effectiveness, mass production for industrial use and good output with better control over dimensions [76,77]. Ag nanomaterials may be generated in various shapes and sizes with great control over the dimensions using the polyol method by changing the parameters, such as concentration of precursor, capping agent, reducing agent, type of solvent, temperature, reaction time and surrounding conditions [78,79]. These nanostructures are mostly obtained via anisotropic growth of the particles, where not all the dimensions of the nanomaterial grow at the same rate. Silver nanorods and nanowires are typical examples of anisotropic growth of nanoparticles in one direction. For AgNWs synthesis at large scale, the polyol method uses silver nitrate as a precursor for the synthesis, PVP as a polymeric capping agent/surfactant and ethylene glycol (EG) as a reducing agent and solvent. Oxidation of silver is spontaneous and a major problem as reported previously [80,81] To avoid oxidation transition metal halide salts such as FeCl_3_, CuCl_2_, cobalt chloride (CoCl_2_), chromium chloride (CrCl_2_), are platinum chloride (PtCl_2_), are used. Where transition metal ions are first reduced by EG and later these metals scavenge the oxygen from silver nanocrystal surface while the Cl ions help in stabilizing the silver ions in such a way that they will be later available for the growth of AgNWs.

The first ever polyol method using EG was performed by Xia et al. in 2002 [82]. They synthesized bicrystalline nanowires of length 50 µm and aspect ratio ~1000, where EG acted as a reducing agent as well as a solvent. The synthesis process comprised two steps: Pt nanoparticles were used as seeds (1st step) for AgNWs growth (2nd step) due to their similar crystal structure, lattice constants and reduction conditions. They proposed that the silver ions start nucleation and unidirectional growth under the influence of PVP as the surfactant, while the unused Pt seeds are separated, as their size is very small. A systematic process of growth of nanowires, as proposed by Xia et al. later in the same year, is shown below [83]. As the method was not cost effective due to the use of platinum seeds, the same group gave an idea about the self-seeding process, where they show that in controlled conditions such as injection rate, continuous magnetic stirring and optimized AgNO_3_/PVP ratio, AgNO_3_ can be reduced to form Ag nanocrystals. These nanocrystals grow to form AgNW with the same size and aspect ratio without Pt seed [84]. It had been reported that an optimized addition of metal salts affects the morphology (shape and size) of the subsequent nanomaterial [85,86]. Addition of Fe^+2^/Fe^+3^ ions in ethylene glycol solution(EG) with PVP can continuously fluctuate between +2 and +3 oxidation state, thus reducing the supersaturation level and reaction rate [87]. Furthermore, addition of Cl^−^ ions in the reaction help in stabilizing Ag^+^ by making AgCl nanocrystals whose solubility is less than solubility of AgNO_3_ in solution. Hence, Ag seeds are formed at a very slow rate and as a result of low solubility, Ag ions are also released at a slower rate in solution [88]. It must be noted that above-mentioned facts slow the reaction rate, which encourages the anisotropic growth of nanowires in a specific direction. The same concept was used to facilitate the growth of AgNWs where Fe^3+^ was reduced to Fe^2+^ by ethylene glycol. The reduced Fe^2+^ ion scavenges the oxygen effectively from the surface of silver nanocrystal to prevent their dissolution [89]. The concept of fluctuating oxidation states lead the group to perform the same experiment in presence of copper chloride (CuCl and CuCl_2_). Copper exists in in two stable states (Cu^1+^ and Cu^2+^), and all the proceeding were quite similar, with a reasonable mechanism of growth process [88]. Detailed growth procedure and influencing factors have been discussed in further detail in the coming sections.

• The growth mechanism of AgNWs

Even though many researchers have experimented with various parameters, some necessary features are primarily important for high quality AgNW synthesis by polyol method. Ensuring a slow reaction rate to slowly achieve the supersaturation state of existing Ag nuclei in the solution is a major focus [90]. The second feature is avoiding the oxidative etching of silver nanostructures by providing any type of oxygen scavenger in the process. Primarily, the process starts with reduction of AgNO_3_ by ethylene glycol to form Ag atoms. The reduction process continues until the supersaturation state of Ag atom is attained after which the nucleation process of the atom starts. In the presence of PVP as a surfactant, slow reduction rate and availability of Cl ions are the main reasons that prevent the Ag nanocrystal aggregation in the solution. Although various strategies for AgNW synthesis have been used, the growth process is still not completely understood [82,83,84]. A number of nanocrystal shapes can be generated in form of single crystal and twinned silver seeds. Although a minor change in kinetics of the reaction may lead to a big change in morphology of nanocrystals [90,91,92]. The desire to inhabit the minimum energy level state of every Ag atoms resulted in majority of the particle shapes being multiply twinned seeds [93]. The single crystals and single twinned further generates nanocubes, bipyramidal, nanospheres, nanotetrahedrals and nanoctahedrals as shown in Figure 5a. The multiply twinned decahedral has been proven to be the most thermodynamically stable shape [94,95]. Xia et al. proposed that for the growth of AgNW, there are two vital factors: PVP interaction with Ag nuclei and multiply twinned particles (MTP) with decahedral shape. They mentioned that a multiple twinned particle with five-fold symmetry has ten {111} facets at the boundary, which was favorable for the minimum energy state of the system, as confirmed by the TEM image of the synthesized nanowires shown in Figure 5c–e. There are five twin boundaries on both sides of the growing dimensions of the wire and the surface energy associated at twin boundaries is highest as compared to other spots. Consequently, most of the Ag atoms are attracted to these twin boundaries and results in selective crystallization. This causes the growth of particle in a specific direction to form the nanowire. They also mentioned that for every {111} facet, there exits {100} plane, which is perpendicular to the longest dimension of nanowire. The interaction between PVP and {111} facet is very less as compared to strong interaction with the {100} plane. This prevents the silver atoms to combine at {100} facets and allows the syndication with {111} facet as shown in Figure 5b [96]. Once the growth process of MTPs starts, the next problem encountered is the oxidative etching of MTP precursors. As MTP are highly vulnerable to oxidative etching due to the defects available in the structure which can be seen in Figure 5f [97]. Hence, the available oxygen molecule species selectively tries to occupy these spaces and this may result in complete etching of MTP from the surface. It must be noted that oxidative etching can be of good use if other shapes such as nanocubes, nanospheres, nanotriangular and nano-bipyramidal are required. However, specifically for AgNWs, oxygen scavengers are required for high quality synthesis. Researchers have been using various metal halide compounds, such as FeCl_3_, CuCl_2_, CoCl_2_, and CrCl_2_. Fe(III) and Cu(II) are the two mostly used oxygen scavengers. When FeCl_3_/CuCl_2_ is injected in EG solution at high temperature, it is reduced from Fe(III)/Cu(II) to Fe(II)/Cu(I) (Figure 5g,h). Now, both the ions cannot be reduced further in the existing condition and eagerly oxidize by scavenging the oxygen from Ag surface to get back to Fe(III)/Cu(II). Successively, the ions can be further reduced by the EG and the process continues to yield high quality nanowires. It is advisable to perform the experiment under noble gas inert conditions because in such conditions, there will be no oxygen available from outside for oxidative etching. However, the presence of halide ions (Cl^−^) are equally important, as they are required for electrostatic stabilization [98,99]. A significant decrease in the available Ag has been reported after addition of Cl^−^ ions due to AgCl formation and a comparatively slow release time of Ag^+^ ions was observed. It should be noted that there are many ways to add Cl^−^ ions, e.g., prior to the AgNO_3_, along with AgNO_3_, and using AgCl as a precursor [100,101,102]. Although many parameter modifications have been studied, the process of successive multiple growth (SMG) is unique. Successive multiple step is used to synthesize the ultralong nanowires, i.e., of length up to 400 µm [102]. The initially formed nanowires are used as seeds to grow longer nanowires in multiple steps. All the conditions resemble the polyol method except the addition of an extra nanowire solution that acts as a precursor. A schematic for successive multiple growth with SEM images is shown in Figure 5i–m.

• Factors effecting the polyol mediated synthesis of NW.

The compact synthesis process of AgNWs with variety of chemical components has led us to define the effects of these components separately. As we have already discussed the growth mechanism and historical development in detail, here, we will discuss the individual parameters that effect the AgNW growth using the polyol process.

a PVP/AgNO_3_ ratio

It has been already reported that a low precursor concentration leads to low chemical potential for nanomaterial crystallization [103,104,105,106]. Researchers have optimized the AgNO_3_ concentration accordingly in polyol process, but no specific concentration can be taken as ideal [32,88,107]. In the early stage of polyol method, Xia et al. noted that 1.5:1 is the best ratio for PVP:AgNO_3_ [84]. Lin et al. varied the silver nitrate concentration by fixing all other parameters and noted that at lower (2:1) PVP (360 K)/Ag NO_3_ molar ratio, AgNWs with wider diameters were formed. However, at a much higher (32:1) PVP/Ag NO_3_ molecular ratio, only nanoparticles were formed, which suggests that excessive use of PVP leads to interaction with all facets in a similar way, as can be seen in Figure 6a–f. The excess amount of PVP promotes interaction between PVP and 111 plane which results in nanoparticle formation Figure 6f. They mentioned that best nanowires were synthesized at a 16:1 ratio [103]. The same approach was used by Zhan et al. to investigate the effect of Ag precursor concentration (100–600 mM) and they noted that the nanowire quantity can be scaled up 10 times if an optimized concentration is used without changing the other parameters [104]. An optimized concentration resulted in high aspect ratio AgNWs (from 27 to 100 times) which can be seen in Figure 6g. Coskun et al. presented a detailed study of AgNWs and they noticed that both injection rate and silver precursor concentration effect the dimensions of the AgNW [101]. They observed that slowing the reaction rate by utilizing a very slow injection rate of Ag precursor could result in high quality nanowires. With increasing injection rate the diameter of nanowire increased, while the nanowire length first decrease and then increase as shown in Figure 6h. They further tuned the PVP/AgNO_3_ ratio and reported that 7.5:1 is the optimum ratio for obtaining long nanowires as shown in Figure 6i. Another study to optimize the AgNO_3_ concentration (25 mM–100 mM) was conducted by Zhang et al. They mentioned that the best ratio of PVP to silver precursor is 2:1 where they obtained nanowires with an aspect ratio of nearly 1000 [108]. As PVP is available in different molecular weight, it has been reported that the chain length of PVP also affects the nanowire dimensions significantly [109,110]. Ran et al. depicted a detailed study with different PVP molecular weights (55 k, 360 k and 1300 k) and obtained different nanowire dimensions ((L = 25 μm, D = 100 nm), (L = 46 μm, D = 60 nm) and (L = 50 μm, D = 80 nm)), respectively [105]. It has also been suggested that the shorter polymer chains will result in shorter and wider nanowires, while long chains can result in longer and narrower nanowires. They also mentioned that when a mixture of PVP-55 k and PVP-360 k was used, nanowires with an aspect ratio more than 1000 could be obtained as shown in Figure 6j–m.

b. Metallic salts

Metal salts are of prime importance for the synthesis of longer nanowires. Various metallic salts have been used for scavenging the oxygen and electrostatic stabilization of Ag ions (Figure 5g,h). Zhang et al. synthesized ultralong nanowires of average length 220 μm and aspect ratio of 4000 times by optimizing the metallic salt (FeCl_3_) concentration as shown in Figure 7e–g [111]. They varied the FeCl_3_ concentration from 0 to 25 μm and found that 12.5 μm is the best concentration for synthesis. As the concentration decreased (<12.5 μm) the diameter of nanowire increased and on increasing the concentration (>12.5 μm) the nanowire length decreased. It has been reported that KBr can also be used as a conucleaent for one-step polyol method because of its good passivation property. Zhang et al. mentioned that KBr cannot grow the nanowires alone, but with NaCl/FeCl_3_ it can considerably help in reducing the nanowire diameter. They synthesized nanowires of sub 30 nm range with KBr assisted conucleation method [112]. The optimized ratio of NaCl:KBr (4:1) was used to synthesize ultralong nanowires with a diameter of 40 nm and an aspect ratio of about 2500; an SEM image of the synthesized nanowires is shown in Figure 7a,b [113]. They mentioned that Br^−^ ions form smaller AgBr crystals than AgCl and passivates the 100 plane to help in diameter reduction. A schematic of this process is shown in Figure 7c. Cu ions are also used for the same contribution in synthesis. In order to investigate the correlation between copper ions concentration and nanowire dimension, Wang et al. demonstrated a rapid (30 min) synthesis polyol process at 160 °C. They demonstrated that an optimized CuCl_2_ concentration (0.36 mM) is necessary for longer nanowires Figure 7d [114].

c. Temperature and time

The scientific measurement of temperature is attributed from average kinetic energy of the particles constrained in the system. One can conclude that temperature plays a crucial role for the interaction between the molecules as a little change in temperature can significantly change the results. Along with that it is well known that at high temperature ethylene glycol changes to glycolaldehyde, which can easily reduce Ag ions to pure Ag atoms [115]. Xia et al. used 160 °C temperature for 1 h in most of the AgNW synthesis. They mentioned that no nanowires were synthesized at 100 °C temperature while at higher temperature (>185 °C) only nanorods were formed. Unalan et al. reported a detailed study on temperature variation from 110 °C to 190 °C and found the optimum temperature is 170 °C [101]. Bergin et al. suggested that low temperature and longer synthesis time can enhance the length of nanowires with a vital difference while a quick synthesis leads to shorter but thinner nanowires [116]. Another report in support of the earlier hypothesis has been provided, where low temperature (130 °C) for longer time (12 h) leads the researcher to obtain nanowires of length 100 μm [117]. As the time is reduced (5 h) the nanowire length reduced to 65 μm at the same temperature

d. Stirring rate

The Ag nanowire synthesis using the polyol method is a one-pot process that does not require any external interception during the synthesis. Even oxygen present in the surrounding can affect the results, which is why most of the time the reaction is conducted in argon or nitrogen atomosphere. Researchers prefer autoclaves and oil baths to keep the reaction undisturbed from external influences [74,111,118,119]. As a proof of concept, Jiu et al. investigated the effect of stirring on nanowire synthesis. AgNWs of length upto 100 µm have been synthesized by adjusting the stirring speed (0–700 rpm) during the process [117]. The nanowires obtained without stirring were three times larger than those with stirring; an SEM image with histogram representation is shown in Figure 7h–k. It has been establish that higher stirring speeds may result in disturbing the reaction by transferring the oxygen molecule. The O_2_ molcecule results in oxidative etching that restricts decahedral twinned seed formation and promotes the formation of other nanostructures. Amirjani et al. studied the effect of stirring (0–2000 rpm) and recommended that high stirring rate results in low aspect ratio of nanowires and nanorods [120]. A slow stirring rate can be used to effectively reduce the AgNW from 200 nm to 130 nm. However, some studies have shown that increasing the stirring rate results in reducing both dimensions (diameter and length) of the nanowire [101].

## 3. AgNW Based Transparent Conductive Electrodes

Transparent conducting electrodes (TCEs) with high optical transmittance and low sheet resistance are crucial for development and performance of optoelectronic devices [128]. ITO is the most commonly used transparent conductive electrode (TCE) for various applications such as solar cells [129], flat panel displays [130], LED [131], OLED [132] and touch screen [133], because of its excellent optoelectronic properties [134] (transparency~90% and sheet resistance~10 Ω/sq). However, there are some major limitations of using ITO which include its brittle nature [135] and high fabrication costs [136]. The growth in demands of modern optoelectronic devices needs to fulfil many crucial characteristics such as solution based processing, low cost, scalable fabrication, excellent optical transparency, low sheet resistance, good mechanical (flexible, stretchable and stress consumable) properties and improved chemical stability [13,16]. Hence, these properties are critically important for application and evaluation of TCF performance using the Figure of merit (**FoM**). **FoM** depends on transmittance (**T**) and sheet resistance $R_{s}$ of the material. Transmittance follows lambert beers law and can be evaluated as
(1)$$T=\;e^{-at}$$
(where “***a***” is the absorption coefficient and “***t***” is the film thickness), while the sheet resistance can be simply calculated by keeping the electrodes at unit distance from each other such as the two or four probe method. The relation between **T** and $R_{s}$ can be written as:(2)$$T={\left(1+\frac{Z_{0}}{2R_{s}}\frac{\sigma_{op}}{\sigma_{DC}}\right)}^{-2}$$
where T is transmittance, ***Z***_0_ is impedance of free space (377 Ω) and the ratio of $\sigma_{DC}$ and $\sigma_{op}$ is known as (FoM), which determines the goodness of the fabricated TCF [137].
(3)$$FoM=\;\frac{\sigma_{DC}}{\sigma_{op}}\;=\frac{377}{\;\left(T^{-1/2}\;-1\right)2R_{s}}$$

Researchers have proposed many alternatives to fulfil the above mentioned properties, such as graphene [138], CNT [139], AgNW [13] and (AZO) [140]. Graphene has high in plain conductivity with excellent optical transparency (~95) and low sheet resistance (50–1000 Ω) [141]. However solution processed graphene has high sheet resistance and chemical vapor deposition technique is too expensive for large scale applications due to the high temperature (>1000 °C) requirement [142]. CNT based transparent conducting electrodes are highly transparent (~90%), but the sheet resistance (200–1000 Ω) [139] of CNT is significantly higher as compared to ITO. Metallic nanostructures especially AgNWs (AgNWs) have arisen as a perfect alternative for transparent conducting electrodes due to their flexibility, low intrinsic resistance and excellent optical transmission [29]. The major requirement of transparency can be achieved by decreasing the density of the solution and minimizing the size of the material at the nanoscale [143]. Nanowires are the ideal material for such applications because of their crosslinking structure as they have two dimensions quantum mechanically confined to the nanoscale and one unconfined dimension that helps in conductivity and crosslinking to each other. Specific applications can be achieved by choosing a suitable nanowire size and diameter.

### 3.1. Transparent Conducting Electrode Fabrication Methods

In order to fabricate a AgNW solution based flexible transparent conducting electrodes (FTCE), diverse methods have been utilized such as spray coating [107,144,145], spin coating, drop casting, dip coating, vacuum filtration and transfer, roll to roll processing, mayer rod coating, doctor blade coating and printing. AgNW based FTCE are easy to fabricate and cost effective with high optical transmittance, high electrical conductivity and flexibility which makes it a probable contender for a range of future optoelectronic devices [31]. Finally a Table 2 has been inserted in the last of Section 3 to compare all TCE fabrication method along with variety of enhancement methods.

#### 3.1.1. Spray Coating

Spray coating is the simplest method of FTEC fabrication. It uses an airbrush or electrostatic sprayer with a very dilute concentration of AgNW solution for coating large areas on a variety of substrates [146,147]. There are many parameters that affect spray coating, e.g., flow rate, pressure of the nozzle, density of the solution sprayed, distance between substrate and nozzle along with the surrounding conditions such as temperature and humidity. Madaria et al. utilized a spray coating technique to obtain transparent conductive AgNW films on random substrates with high uniformity. The optical transparency and sheet resistance of the films could be optimized by varying the density of the AgNWs in the spraying solution. Using a polydimethylsiloxane (PDMS) assisted contact transfer technique, they transfer the AgNW network to PET substrates to fabricate a flexible TCE with an optical transparency of 85% and sheet resistance of 33 Ω/sq [148]. The biggest challenges for fabricating flexible TCE is to transfer the electrode to dissimilar arbitrary substrates because of the risk of cracking and damaging [146,149]. Consequently, a transparent conducting AgNW–polydimethylsiloxane (PDMS) electrode fabrication and transfer technique by water-assisted transfer printing method was reported as shown in Figure 8h [149]. This advanced method displays a strong capability for transferring the AgNW film on various hydrophilic substrates without any significant change in the characteristics of the film as depicted in Figure 8i. A high ratio of DC conductivity ($\sigma_{DC}$) to optical conductivity ($\sigma_{op}$) with 82% transmittance, 9 Ω/sq sheet resistance, tensile strain (0 to 50%), and 2 mm bending radius flexibility was obtained without significant loss of conductivity. Choi and coworkers demonstrated a two-step spray-coating method for fabrication of AgNW–PEDOT:PSS composite based flexible electrode as shown in Figure 8a–e. The first step involves spray coating AgNW and later PEDOT:PSS is spray coated on predeposited AgNW to attain 84.3% transmittance and 10.76 Ω/sq sheet resistance [144]. The PEDOT:PSS significantly enhances the stability under compression and bending, reduces the junction resistance and surface roughness as shown in Figure 8f,g.

The correlation between T and R_s_ can be evaluated by percolative FoM and the percolation exponent [150]. The percolative FoM depends on nanowire properties, while the percolation exponent depends on the properties of the network. Hence, two strategies can be implemented in order to enhance the output of TCEs. The former consists of enhancing the nanowire dimensions (high aspect ratio), and the latter of improving the network properties (low junction resistance and higher stability). Scardaci et al. tried to improve the network properties by optimizing the percolation exponent and found that nozzle pressure is the critical parameter that effect the size of the droplets, network uniformity and the morphology of the fabricated film [151]. They fabricated a TCE with 90% transmittance and 50 Ω/sq sheet resistance by varying the nozzle pressure while a maximum 94% transparency was achieved at higher sheet resistance. In 2020, another group of researchers tried to enhance the percolative FoM by refining the dimensions of nanowires (average length 130 µm, aspect ratio > 1000) and the synthesized AgNWs were spray coated on a glass substrate to obtain a TCE of 91% transmittance and 4.6 Ω/sq sheet resistance [122].

#### 3.1.2. Spin Coating

Spin coating is considered as the most common and simple way for AgNW based FTCE fabrication [152,153,154,155]. During spin coating, a low concentration of AgNW solution is dropped on the substrate that is spun by the spin coater under a high centrifugal force, resulting in the solution being uniformly distributed over the substrate. There are different parameters that affect the properties of the fabricated TCE such as spinning rate, concentration of the solution and the time duration of coating. Leem and coworkers used an oxygen plasma pretreated glass substrate to optimize spin coating parameters by varying the spinning rate in the range 1000–5000 rpm. Nanowires with a length of several tens of micrometers were used and they found that the transmittance of the film decreased significantly when the spinning rate decreased from 5000 rpm (98% transmittance with 36 Ω/sq sheet resistance) to 1000 rpm (89% transmittance with 10 Ω/sq sheet resistance), which can be seen in Figure 9e below [153].

Apart from the mentioned parameters, the nanowire dimensions such as very thin diameter (~20 nm) and high aspect ratio (>1000) are the two most notable characteristics to enhance percolative FoMs. Selective precipitation method was used for nanowire purification to enhance the transparency by 2% and finally a TCE with 97.5% transmittance and 70 Ω/sq sheet resistance was fabricated by spin coating followed by thermal annealing [155]. AgNW based conducting films have non-uniformity and high surface roughness due to random alignment and junctions between nanowires. A composite PVA-AgNW film of 86.9% transmittance and 0.75 Ω/sq sheet resistance was fabricated to improve the mentioned properties as shown in AFM image of bare and composite film in Figure 9b–c [156]. Two separate PET substrates were used to fabricate AgNW and PVA film. Consequently, the fabricated AgNW film was positioned on the PVA film to completely cover the AgNW network using mechanical pressure; a schematic describing the procedure for film fabrication is shown in Figure 9a, and a transmission-resistance graph is presented in Figure 9d. AgNWs have been known to suffer from oxidation when exposed to chemicals or air. It has been reported that amorphous aluminium doped zinc oxide (a-AZO) has a highly compact structure, which can effectively guard the AgNW network to enhance the chemical stability as an anti-oxidative shield. An optimized three layered a-AZO/AgNW/ AZO composite electrode was fabricated using spin coating with a 88.6% transmittance and 11.86 Ω/sq sheet resistance [157].

#### 3.1.3. Drop Casting

Drop casting comprises of dropping a dilute AgNW dispersed solution on a transparent substrate. Usually a surface treatment of the substrate is required before drop casting to ensure that the solution evenly spreads on the substrate surface [158,159]. If pretreatment is not performed, there are higher chances of nonuniform distribution of the AgNWs on the surface, resulting in increased surface roughness.

Generally, a thermal treatment is preferred after the solution is drop casted on surface to remove the dispersant and enable annealing of the AgNW network. The AgNW solution concentration and volume of the dropped solution are two important factors that must be optimized to fabricate a good transparent electrode using drop casting. A flexible TCE made of poly-acrylate (PA) AgNW composite with high transparency and low sheet resistance comparable to flexible ITO electrodes was fabricated using drop casting as shown in Figure 10a followed by curing with UV light to enhance the electrode properties [160]. Cheng et al. synthesized AgNWs with great aspect ratio via a novel polyol method without the use of halides ions [161]. The as-prepared AgNWs were drop casted on a patterned glass substrate to form a uniform conductive network. After drying, liquid polydimethylsiloxane (PDMS) was poured on the AgNW network layer followed by curing and peeling off the AgNW/PDMS film. This stretchable electrode displayed good optoelectronic characteristics with a sheet resistance of 14 Ω/sq and a transmittance of 90% without any post treatment as shown in Figure 10c,d. A schematic of the fabrication procedure been shown in Figure 10b. In 2019, Chen and coworkers demonstrated a modified drop casting method to fabricate AgNW-PDMS composite based flexible transparent conducting film, which fulfils some of the key requirements of modern optoelectronics [162]. The AgNW network was drop casted on the PDMS substrate to create the required pattern with the help of a stencil mask. The PDMS substrate was pretreated with oxygen plasma to enhance the adhesive properties between the AgNW and the substrate. The conductive polymer film displayed good optoelectronic properties with a 15 Ω/sq sheet resistance, 85% transmittance, a relatively uniform surface with surface roughness of about ∼22 nm along with 70% stretchability, and 70% bending which have been shown in Figure 10e–g. It is worth mentioning that drop-casting technique is not highly scalable and the process is not completely repeatable each time.

#### 3.1.4. Vacuum Filtration and Transfer

The vacuum filtration method is quite popular for its usability for transferring AgNWs on a variety of substrates including polymer films [163,164,165,166,167,168]. The process includes pouring or dropping a dilute AgNW solution on a filter paper supported by a vacuum filtration. The filter paper helps in constructing the nanowires network uniformly by filtering out the solvent and nanocrystals. Consequently, high quality nanowires network can be obtained on the filter paper, which can then be transferred to another substrate. The filter paper pore size, vacuum pressure and AgNW solution concentration [168] are the parameters of prime importance. Wu et al. for the first time introduced 1-dimentional nanostructured thin film fabrication using vacuum filtration and transfer [163]. Though the aspect ratio of AgNWs was quite low (~76.4), transmittance of 92% and 1 Ω/sq sheet resistance could be achieved. Lee et al. synthesized ultralong nanowires of length up to 500 µm using a successive multiple growth (SMG) method. They fabricated TCEs with a 89–95% transmittance and sheet resistance 9–69 Ω/sq by using vacuum filtration [164]. Hong and coworkers gave a novel idea of using cellulose nanofiber paper that acts as a filter as well as substrate [166]. A schematic of the method with fabricated TCEs is shown in Figure 11a,g as the process avoided the AgNW network transfer to another substrate, a relatively lower 12 Ω/sq sheet resistance with 88% transmittance was observed as compared to other conventional methods shown in Figure 11b. The ‘filtration coating’ offers homogeneity on the surface and low AgNW junction resistance which can be seen in Figure 11c–f. Surface homogeneity was attained as the nanowires are embedded in the pores of the nanofiber paper during the vacuum filtration process. Conventional methods such as spin coating, spray coating and drop casting suffer from the coffee-ring effect since these methods generally use annealing for drying process, which promotes the nanowire agglomeration at specific areas and results in a nonhomogeneous distribution. As the AgNW network shows effective bonding by rooting inside nanofiber paper, it enables good flexibility of the transparent conducting film with negligible changes in the electrical conductivity. Recently, Won et al. introduced a novel and effective fabrication method for patterning a TCE into various shapes by introducing the Kirigami approach for highly flexible and stretchable E-skin applications. These Kirigami patterned electrodes possess ultrastrechability for a range up to 400%, with strain-invariant electrical properties for more than 10,000 cycles [119]. A dilute AgNWs solution was used to transfer the AgNW network onto a glass substrate by vacuum filtration. Next, colorless polyamide (cPI) was spin coated followed by curing (to enhance the adhesion between AgNW and cPI) and peeling off. The fabricated FTCE on cPI substrate exhibited a transmittance of 95% and a sheet resistance of 50 Ω/sq with excellent flexibility and stretchability. The results of a 0 to 400% stretchability test for 10,000 cycles are shown in Figure 11h.

It should be noted that the vacuum filtration method works efficiently for nanocrystals of size higher than the micrometer range. However, while dealing with size dimensions in the nanometer range, the loss of material in the filtrate can be significant and this loss is enhanced during subsequent transfer to another substrate.

#### 3.1.5. Doctor Blade Coating

Doctor blade coating is a simple process that uses a well-dispersed and homogenous nanomaterial ink with suitable solvent, binder and dispersants [169,170,171]. The film thickness depends on the distance between the blade and substrate, while the surface morphology depends on the density of ink and blade speed [172]. Krantz et al. in 2011 reported on AgNW solution processed electrodes as a substitute for ITO for organic solar cell application [170]. AgNWs were deposited on a glass substrate to fabricate transparent conducting electrodes with high transmittance, i.e., 90%, and low sheet resistance, i.e., 9 Ω/sq, by using doctor blade coating at moderate temperatures. Zhang et al. introduced a solution processed multilayer fabrication of sandwiched rGO/AgNWs/rGO TCE on PET and glass at low temperature (~50 °C) [173]. The rGO sheet aids in enhancing electrical contact of AgNWs by connecting detached nanowires and reduces the overall surface roughness. The fabricated film exhibited high transmittance, low sheet resistance and high stability due to presence of rGO as an antioxidative shield. The same research group presented another multilayer film structure consisting of AgNW-poly vinyl butyral (PVB) and aluminum doped ZnO nanoparticles (AZO) using doctor blade coating method and attained a very low sheet resistance with a transmittance of over 94%. The PVB layer effectively improves the adhesion property between the glass and film while the AZO NPs enhance the thermal stability and reduce the sheet resistance [174]. Hwang and group introduced a novel material, UV-curable Raygloss, which can be used as a coating layer as shown schematically in Figure 12a. This layer does not significantly affect the optoelectronic properties of the film but considerably enhances the mechanical properties against scratches (100 times shown in Figure 12b), wipes (60 times, shown in Figure 12c), bending (300,000 times) and stability against oxygen [174]. Interestingly, a small increase in optical transparency from ~87.7% to ~88.3% was observed due to the presence of this coating layer. This observation can be explained by Fresnel’s theory of reflection, which states that light reflection at a specific surface can be reduced by introducing a coating layer which has a refractive index value (Raygloss = 1.54) between that of the rarer (air = 1) and denser medium (PET = 1.64) [175].

A highly transparent multilayer structure for a solar cell device embedded with both electron and hole collecting AgNW electrodes was reported by using the doctor blade technique [169]. In 2020, Huang et al. fabricated a polymer based ultraflexible AgNW transparent conducting film on glass and PET substrate. The fabricated TCF properties were optimized by changing the concentration of the AgNWs solution and the height of the blade. The coating speed is considered as a vital parameter for uniform surface coating and. they found that a nonhomogeneous coating of AgNW was observed when the coating speed was less than 50 mm/s. Subsequently, a transparent polymer layer was coated on the AgNW network followed by curing and peeling off from the glass substrate. The polymer film with good optoelectronic properties was highly flexible and could be bent up to 180° without any change in resistance and shows potential for applicability in next-generation flexible and foldable optoelectronic devices [176]. A picture of flat and 180° folded transparent electrode is shown in Figure 12d,e.

#### 3.1.6. Mayer Rod Coating

Mayer rod coating, unlike other solution-processed techniques, is considered as a more scalable [177], accurate and faster method for deposition of nanowires to make a thin film for practical applications. The Mayer-rod consist of a solid bar with helical metal wires wound on it. The ink density, surface of substrate, meniscus height (distance between the substrate and rod) and slit size are the key parameters for Mayer rod coating. Slit size of Mayer rod depends upon the density of wire looping and it can be varied accordingly with application requirements (51–381 μm as shown in Figure 13a) [178]. The process involves positioning the AgNW solution/ink on one side of substrate and then uniformly spreading the solution on the substrate using the Mayer rod [179,180,181]. Hu et al. first fabricated a TCE of 80% transmittance and 20 Ω/sq sheet resistance by Mayer rod coating and later researchers improved the efficiency of the method using minor modifications [182]. A simple and efficient Mayer rod coating method was used for fabrication of highly cross-aligned AgNW transparent electrodes with good optoelectronic properties (21 Ω/sq sheet resistance at 95.0% transmittance) on a large surface area (20 × 20 cm^2^), while the randomly aligned AgNW network provided 90.4% optical transmittance and 21 Ω/sq sheet resistance [180]. An optical image of the aligned nanowires can be seen in Figure 13c,d. The meniscus height and slit size optimization is important as the alig nment pattern of nanowires majorly depends upon the shear force, which further depends on the height of the deposited meniscus film. As shown in Figure 13e, a strong shear force is obtained with a thin meniscus film which helps to align the AgNW in a unidirectional manner. A thick meniscus film suffers from misalig nment due to weak shear force near the substrate surface. A schematic for the procedure is shown in Figure 13b. Yang et al. fabricated an AgNW-PEDOT: PSS composite TCE with a sheet resistance of 12 Ω/sq, transmittance of 96% and a uniform surface with only 7.13 nm surface roughness [183]. The PEDOT: PSS nanosheets aid in welding the nanowire junction to reduce the junction resistance from 32 Ω/sq to 12 Ω/sq and improve the adhesion between AgNWs and the substrate. A SEM image of AgNW-PEDOT:PSS film is shown in Figure 13h. The fabricated TCE was used as the top and bottom electrode for developing highly sensitive and flexible touch panels of 7 × 7 cm^2^ area, which is highly stable against wiping (500 times), bending (5000 times) and writing (2000 times). In 2020, an optimized and quick synthesis of ultralong AgNWs have suggested that Mayer rod coating delivers improved outcomes than spray coating with a significant difference of 3% in transparency (rod coating = 94% and spray = 91%) for same sheet resistance of 5 Ω/sq [122]. The reason for the difference in the results were essentially because of alig nment factor as the rod coated TCE had more aligned nanowires while the spray coated TCE had more curved nanowires as shown in Figure 13f,g. The AgNW based TCE provides an additional advantage of high transparency in NIR region (94%), while the transparency for NIR region decreases to 35% in case of ITO as shown in Figure 13i [184]. Some modern applications demand transparency in the NIR region and AgNW based TCEs show promise for these applications. The change in transparency over a wide wavelength range (*λ* = 0–2500 nm) for AgNW based thin films was quite small.

#### 3.1.7. Roll-to-Roll Slot Die Coating

Roll-to-roll slot die coating is a simple technique for large-scale film deposition on a substrate. The major advantage associated with this method is the large area surface uniformity in a controlled manner with film thickness ranging from the nanometer to micrometer range. Some vital parameters for roll-to-roll processing include ink feeding rate, coated-web gap and web speed. An optimized combination of these parameters can be used to fabricate TCE with high FoM. In order to make easy understanding of forthcoming parameters Figure 14a has been inserted. An increase in the web speed results in thinner coating layer with less density of conducting material which in turn increases the sheet resistance and transmittance [185]. Furthermore, research has shown that silver inks demonstrate better optoelectronic properties as compared to conductive polymer inks. Linda et al. used similar roll-to-roll parameters (web gab = 40 µm, pump rate = 145%, and web speed = 2.4 m/min) and found that the conductive polymer had 100 Ω/sq sheet resistance at 80% transmittance while the AgNW ink had 40 Ω/sq sheet resistance at the same transmittance [186]. An optimized one-step AgNW/ PEDOT:PSS ink based TCE using roll-to-roll slot die coating has also been demonstrated as shown in Figure 14b. AgNW/ PEDOT:PSS was deposited at 2.7 m min^−1^ of ink feeding rate and a static 100 µm shim gap on a large surface area (46 cm × 20 m) PET substrate with good optoelectronic properties (sheet resistance ~75 Ω/sq and transmittance > 90%) [187]. A picture of the fabricated film is shown in Figure 14d. Jeong et al. suggested that calendaring is an impactful process in order to improve the efficiency of roll-to-roll slot die coating method [188]. A combination of roll-to-roll slot die coating and calendaring process machine is shown in Figure 14c which can control roll temperature, pressure and web speed to provide low sheet resistance (28 Ω/sq) with good transmittance (86%). The calendaring process parameters (pressure and temperature) significantly improves the contact between Ag nanowires by eliminating the PVP surfactant coating to reduce the sheet resistance and surface roughness as seen in Figure 14e,f. Kim and coworkers introduced an AgNW network based TCE fabrication on PET substrate using roll-to-roll slot die coating and they achieved varying transmittance (89–90%), haze (0.5–1%) and sheet resistance (30–70 Ω/sq) by optimizing the flow rate of shear thinning silver ink in die slot. The final application as a touch screen has 90% optical transmittance and sheet resistance of 50 Ω/sq [189].

#### 3.1.8. Dip-Coating

Dip-coating is a very simple method of dipping the substrate in a solution containing the nanomaterial to be coated. After dip-coating, the substrate is removed from the solution followed by a drying process. During the drying/annealing process, the nanowires stay on the surface while the solvent is evaporated. Solution concentration, withdrawing speed of the substrate and atmospheric drying conditions (temperature and pressure) are critical parameters, which must be optimized in order to attain good optoelectronic properties of the TCE [190]. The dipping direction of the substrate can be changed to reduce the sheet resistance and increase the number of nanowire junctions [191]. Ackermann et al. reported a low haze and ultra-TCE with a sheet resistance less than 100 Ω/sq by simply optimizing the concentration of the AgNW, the substrate dipping direction and withdrawing speed [192]. The improved optoelectronic performance can be seen from the transmittance vs. resistance graph as shown in Figure 15a. In order to further enhance the optoelectronic properties, a two-step dip coating method on PET substrates can also be used [193], as demonstrated in Figure 15b,c. The flexible PET film was dipped into the AgNW solution (0.1 mg/mL) and withdrawn at a speed of 1 cm s^−1^ with the help of a programmed machine. In the second step, the substrate was rotated by 90° and dipped to enhance the number of nanowire junctions and improve conductivity. The two step dip-coating resulted in a fabricated TCE with improved optoelectronic properties (T = 92% and R_s_ = 35 Ω/sq). Similarly a simple method of homogeneous AgNW network electrode on glass has been fabricated by recognizing suitable processing constraints such as dipping cycles (between 10 and 150 cycles) and withdrawing speeds (0.2–2.5 mm/s) [194]. A further treatment of annealing and PEDOT:PSS coating is used to reduce the surface roughness and obtain 85% transmittance and 15 Ω/sq sheet resistance. Ahn et al. presented a detailed discussion on optoelectronic properties of AgNW thin film using dip coating for mass production [195]. A quality correlation between the nanowire parameters such as nanowire length, diameter and aspect ratio has been explained.

#### 3.1.9. Printing Techniques

Printing techniques such as inkjet printing [196,197], gravure printing, screen printing, electro-hydrodynamic jet (EHD) printing and capillary printing have been widely studied, as these techniques enable the printing of flexible and stretchable electrodes. In order to attain a TCE with a high figure of merit, the selection of ink dispersant and concentration of the AgNW must be optimized to obtain a suitable ink. Printing techniques have been widely utilized for large scale production of TCEs, but the minimum achievable resolutions are still in the low micrometer range. It is worth mentioning that substrate and ink interaction, printing speed and ink pressure play a vital role in effecting the quality and printing accuracy of the ink.

Lu et al. used an inkjet printing technique to fabricate the upper electrode of a semi-transparent organic photovoltaic (OPV) device with a power efficiency of 2.71% [196]. They observed that printing the ink in successive steps could immensely enhance the conductivity while resulting in a slight decrease in transparency of the electrodes. Li et al. used a screen printing technique to fabricate flexible and stretchable TCE with a transmittance of 73% and sheet resistance less than 2 Ω/sq [198]. A high FoM (135) with good stability after 1000 bending and stretching cycles was achieved by optimizing the printing parameters, ink formulation and laser post treatment. Recently, a AgNW/GO hybrid TCE with enhanced optoelectronic properties was fabricated using screen printing and showed feasibility for long-term electrocardiography monitoring. The AgNW dispersion was printed on a PET surface followed by coating of a GO layer to avoid oxidation [199]. Gravure printing involves the use of an engraved cylinder which is fixed to a rotary printing press and has been commonly used for commercial printing of magazines and newspapers, [5,200]. The importance and relation between ink pressure, speed and rheological behavior of AgNW ink during gravure printing has been reported [201]. Gravure printing has been implemented to fabricate a AgNW based TCE on PET substrate with 95% transmittance and 32 Ω/sq sheet resistance. In order to attain accuracy and control over percolative FoM and the percolation exponent, capillary printing techniques have also been introduced [202]. It was observed that the partial alignment of AgNWs during capillary printing shows better optoelectronic properties as compared to randomly aligned AgNWs. Furthermore, the luminance efficiency of the fabricated polymer light-emitting diodes (PLEDs) increased by 30% for partially aligned AgNWs. In addition, a polymer solar cell (PSC) with a power efficiency of 8.57% was also fabricated, which is one of the highest reported values obtained using AgNW based TCE. All the above mentioned printing techniques are still limited in terms of the minimum achievable resolution [203]. It is also worth mentioning that EHD printing was shown to be able to approach a higher resolution and accuracy as compared to other techniques [204]. A detailed study to fabricate flexible TCE and Ag grids of different line width and line gap has been demonstrated using EHD printing [205]. Park et al. used a modified electro-aerodynamic jet printing technique to implement an aerosol deposition method in order to enhance the optoelectronic properties of the fabricated Ag grid [206]. Ag nanoparticles were electrically charged and sprayed under application of a high AC voltage (7 kV) on a prepatterned Ag grid based PET substrate. Consequently, a reduction in sheet resistance from 7.38 Ω/sq to 1.95 Ω/sq was observed at 84% transmittance. Cui et al. reported AgNW based EHD printing techniques for large area substrates and demonstrated a dependence of ink viscosity, AgNW concentration and blade moving speed on the final results obtained on a variety of substrates such as PDMS, PET, Paper and glass [207]. We anticipate that the performance of these printing techniques will continue to improve in the future and enable large scale industrial TCE manufacturing for a wide range of optoelectronics devices.

### 3.2. TCE Enhancement Method

A range of treatment methods have been utilized to reduce the contact resistance (R_c_) at the nanowire-nanowire overlapping junction and improve the optoelectronic properties of the solution processed TCE. The polyol method, which is the most commonly used method for large-scale synthesis of AgNWs, generally uses PVP as a surfactant to avoid agglomeration between AgNWs. For e.g., a 50 nm PVP coating as shown in Figure 16a can result in high junction resistance and pretreatment is required to reduce the thickness of the PVP layer as shown in Figure 16b,c. A range of pre and post fabrication treatment methods such as solvent washing, mechanical pressing, joule heating, thermal annealing, plasmonic welding, polymerization and electro-deposition have been used by researchers to improve the final optoelectronic properties of the TCE as discussed in further detail in the following sections.

#### 3.2.1. Solvent Washing

Solvent washing, also known as centrifugation technique, is a widely used pretreatment method after AgNW synthesis to separate out organic and inorganic contaminants. In a solution based AgNW TCE, the value of the sheet resistance (R_S_) primarily depends on the nanowire contact resistance (R_c_). The PVP encapsulating layer insulates the transmission of electrons at the nanowire junction by quantum tunneling effects as shown in Figure 17a. Consequently, the PVP layer, which can act as a tunneling barrier, can be removed or its thickness can be reduced to decrease the R_c_ as well as the sheet resistance (R_S_). It has been demonstrated that with multiple washing process with water there is a significant reduction in R_c_ (Figure 17b). As a proof of the concept, TEM images of AgNW without washing and after washing for 20 times have been shown in Figure 17c [208]. A significant reduction in the thickness of the PVP layer by about 75% was observed after the washing step. Furthermore, two transparent conducting electrodes were fabricated using Mayer rod to investigate the difference in optoelectronic properties. Improvement in transmittance and reduction in haze were seen after multiple washing steps, as shown in Figure 17d. The removal of PVP layer using a post-treatment method after TCE fabrication typically restricts the final application of the device due to substrate susceptibility toward various parameters such as solvent used and treatment time. Wang et al. performed a pretreatment of AgNW by solvent washing (ethanol) and observed a significant decrease in PVP layer thickness from 13.19 nm to 0.96 nm after four solvent washing steps [209]. A dramatic decrease in sheet resistance from 5 × 10^4^ Ω/sq to 93 Ω/sq at 95% transmittance was observed after the washing steps. Lee and coworkers used methanol to remove the PVP layer to efficiently enhance the nanowire junction conductivity by reducing the PVP layer thickness from 4 nm to 0.5 nm using 5 successive washing cycles [210].

#### 3.2.2. Thermal Annealing

Thermal annealing has been widely used as a post treatment method for AgNW based TCEs due to its simplicity, feasibility and efficiency at large scale [106,153,164,211,212]. The temperature annealing process can be implemented in two ways, which include low temperature annealing (<150 °C) for longer time and high temperature (~200 °C) annealing for shorter time depending upon the nanowire and substrate stability. Zhang et al. fabricated a highly stable and uniform TCF using silver ink dispersed in ethanol via Mayer rod coating [213]. Various post treatment methods such as natural drying, static heating and dynamic heating (as shown in Figure 18a–c, respectively) were used to investigate the optoelectronic properties and surface morphology. Sheet resistance of 24 Ω/sq and 91% transmittance with excellent surface uniformity over a large area (20 × 40 cm) were accomplished in case of dynamic heating (UF = 0.2915) treatment as compared to natural drying (UF = 0.5167) and static heating (UF = 0.3415). Based on the PVP layer coating, it is important to choose an optimized temperature for annealing. Thermal analysis of PVP indicates that the glass transition temperature (174 °C) and melting point (224 °C) are quite high and the substrate material can be susceptible at these temperatures. Air dried AgNW TCE was annealed at 200 °C and this resulted in slight melting of the PVP, which aids in nanowire fusion [214]. An SEM image of the melted PVP and nanowire fusion is shown in Figure 18d,e. A detailed model on the evolution of AgNW sheet resistance (reduction, stabilization and spheroidisation of AgNWs which can be seen in Figure 18f–j) by thermal annealing has been reported [215]. A TCE of 90% transmittance and 9.5 Ω/sq of R_s_ was fabricated by optimizing the annealing conditions. Four possible mechanisms for resistance variation with temperature which include Rayleigh instability of the AgNW diameter, organic residue desorption, PVP degradation and sintering have been mentioned. Later, the same group proposed a method to investigate the optoelectronic properties of AgNW based TCE by optimizing nanowire density, dimensions and thermal annealing procedure [216]. They proposed that thermal stability is dependent on the nanowire size dimensions and suitable selection of all three parameters results in highest reported FoM with T = 89.2% and R_s_ of 2.9 Ω/sq. For organic applications, humidity assisted annealing is a better approach, which requires a low temperature for R_s_ reduction. The humidity absorption at the surface of PVP reduces the glass transition temperature by softening it, which enables annealing to be performed at a lower temperature of 60 °C [217]. The method is quite efficient as annealing at 60 °C provides the same results with conventionally annealed TCEs at ~200 °C.

#### 3.2.3. Mechanical Pressing

Many substrates are susceptible to thermal annealing at high temperatures [212,218,219,220], and mechanical pressing can be used as an alternative when low temperatures are more suitable for TCE fabrication [165,221,222]. Mechanical pressing is a simple technique whereby pressure is applied to enhance the junction conductivity of the AgNW network. Mechanical pressure in the Mega Pascal (MPa) range is commonly used for short time durations. Some researchers also implement hot pressing to control the humidity and temperature [222]. Hot pressing with two rollers at 5 mm/sec was performed three times to enhance the electrical properties of the TCE [223]. Hu and coworkers applied an extremely high mechanical pressure of 81GPa for 50 s to investigate the enhancement in optoelectronic properties. A significant reduction in surface roughness from 110 to 47 nm and R_s_ by 10 times was observed [182]. Tokuno and coworkers demonstrated that mechanical pressing could enhance optoelectronic properties of heat sensitive materials [214]. They applied 25MPa of mechanical pressure for a time span of 5 s to attain a sheet resistance of 8.6 Ω/sq and 80% transmittance on PET substrates, which cannot be heated at 200 °C. Furthermore, they observed a significant reduction (one-third) in surface roughness as compared to the heat treated electrode. The enhancement in NW-NW junction can be seen from FE-SEM images and the resulting R_s_ vs. pressure graphs are shown in Figure 19a–c. Du et al. demonstrated that after mechanical pressing R_s_ can be reduced by one order from 172.89 Ω/sq to 18.6 Ω/sq using a mechanical pressure of 25 MPa [224]. In addition, a sharp contrast in I–V curve characteristics after mechanical pressing was observed as shown in Figure 19d. A further treatment of the AgNW network with UV radiation results in TCE with high FoM (1.35 × 10^−2^ Ω^−1^). Mechanical pressing technique can also be useful in fabricating AgNW based TCFs. He et al. fabricated a PVA-AgNW based TCF using mechanical pressing [156]. The PVA film was first softened by IR radiation followed by inverting the AgNW coated PET onto the PVA film under the application of 30 MPa mechanical pressure for 4 min. A resulting R_s_ of ~1 Ω/sq at 85% transmittance was reported (Figure 19e,f). A significant increase in the nanowire diameter was observed after mechanically pressing as shown in Figure 19g,h.

#### 3.2.4. Cold Welding

Due to weak metallic bonding, it is quite easy to directly weld metal nanowires by external treatments [225,226]. Although AgNWs have good optoelectronic properties, the presence of an induced nanoscale gap can result in poor NW-NW and NW-substrate contact. Capillarity force can be a good choice while dealing at the nanoscale range [227,228,229]. There have been many publications about using capillarity force as an assembler of nanostructure designs from namomaterials (NPs and NWs) [230,231]. It is worth mentioning that on the nanometer scale, the pulling force of a liquid bridge between two nanowires can reach up to the mega newton range. A simple and quick method of nanowire welding at room temperature has been reported by applying moisture by breathing or water mist [232]. Capillarity force induced by breathing or water mist is an impactful approach for nanowire welding under normal conditions for 30–40 s without annealing as shown in Figure 20h. A negligible variation in transparency with a momentous drop in resistance (from 2.25 × 10^5^ Ω/sq to179 Ω/sq and 6.3 × 10^4^ Ω/sq to 37 Ω/sq) was observed after successive moisture treatments. A schematic for the procedure and observed changes have been shown in Figure 20a–g. Xu et al. synthesized very long nanowires with a maximum length of 200 µm and utilized these nanowires to fabricate a TCE with R_s_ = 8.6 Ω/sq and T > 91% without any post treatment method. Later they demonstrated a method of AgNW cold welding by dipping the fabricated electrode in CTAB solution for enhancement of transmittance (from 91.3 to 93%) and high FoM [125]. A small decrease in R_s_ value by scavenging PVP layer was observed by CTAB solution using SERS spectra while the enhanced optoelectronic properties can be seen in Figure 20i. It has also been reported that there is an exceptional wedge-shaped nanoscale gap between the AgNW and substrate, as shown in Figure 20j. This NW-substrate gap plays an important role in the overall R_s_. A smart selection of substrate and welding element can improve the results further by improving NW-substrate contact. Zhang et al. used DI water to enhance both NW-NW and NW-substrate contact and demonstrated that a hydrophilic substrate was more efficient as compared to hydrophobic substrate. The capillarity force in case of hydrophilic substrate was more and better NW-substrate contact was obtained, which resulted in a decrease in sheet resistance by 2500 times [233]. A schematic illustration of this mechanism is shown in Figure 20k for better understanding.

#### 3.2.5. Joule Heating and Electro-Deposition Methods

Random arrangement and high surface roughness are the two major drawbacks of nanowire thin films. Recent reports suggested that joule heating by either electric sintering or plasma treatment can enhance the welding of junctions between nanowires [234,235]. Yan and coworkers fabricated a TCE by spray coating AgNWs followed by drop casting GO to enhance the electrical conductivity between nanowires [236]. They performed a post-treatment by using an electrical sintering process which comprised of applying a DC voltage across the TCE to fuse the nanowires at the junction sites which resulted in a decrease in R_s_ from 1 KΩ/sq to 60 Ω/sq at 86% transmittance. Furthermore, an optimized electric field can also be used to align AgNWs to obtain a TCE with good optoelectronic properties. As a proof of concept, AgNW based TCEs were exposed to different DC voltages to investigate the effect of electric annealing. An abrupt decrease in sheet resistance was observed due to activation of efficient percolating pathways (EPPs). These percolation pathways, as observed under IR surveillance, increased with increasing DC voltage as shown in Figure 21a and resulted in improved contact at the nanowire junction [237].

Recently, a highly flexible, transparent and conductive electrode with a negligible junction height was fabricated using spray coating under the application of low DC voltage (3.6–6.0 V) as shown in Figure 21b [238]. By utilizing an optimized DC voltage, spray coating time and plasma treatment, they obtained a FoM of 5.52 × 10^−2^ Ω^−1^ with a R_s_ of 4.64 Ω/sq and 87.3% transmittance. A significant reduction in surface roughness from 16.83 nm to 13 nm was seen after the application of an electric field as shown in Figure 21c–d. Nguyen et al. also used joule heating to reduce the surface roughness of the fabricated TCE to only 1.92 nm and reported a significant decrease of 19% in sheet resistance (9.9 Ω/sq) at 92% transmittance [239]. Later they fabricated an AgNW embedded polyamide composite film by annealing and hot pressing. The ultrasmooth, highly conductive and flexible film peeled from the substrate as illustrated in the schematic in Figure 21e. Another useful post-treatment technique that can be utilized to improve the optoelectronic properties of AgNW based TCE is electrodeposition. In 2019, Lee et.al. used electrodeposition of silver ions on spin coated AgNW TCE and demonstrated that this technique can enhance various optoelectronic properties such as electrical conductivity, optical transmittance, high temperature stability along with low junction height and sheet resistance [240]. Consequently, the obtained a TCE with a transmittance more than 93% and sheet resistance of 14 Ω/sq as shown in Figure 21f,g.

#### 3.2.6. Nanowelding by Chemicals and Polymers

Research studies have also utilized a range of chemicals and polymers to perform nanowelding and enhance the NW-NW junction and NW-substrate contact [23,241,242]. Lu et al. demonstrated alcohol based precursor solution method to create and grow silver “solders” for welding AgNW junctions in order to reduce the junction sheet resistance and enhance stability of the TCE [243]. Briefly, a precursor solution of silver nitrate (AgNO_3_), nitric acid (HNO_3_) and ascorbic acid (AA) with a pH of about 3 was prepared in ethanol to use as precursor which completely hinder the reaction between AgNO_3_ and AA. The fabricated TCE was treated with this precursor solution which induces the capillarity force to enhance the NW-NW junctions as shown in Figure 22a–c. This simple post-treatment method does not require optimization of external parameters such as temperature, humidity, electric current and mechanical pressure and can be performed in normal ambient surrounding. Another effective method of enhancing NW-NW junction is by using hydrogen chloride vapor in the presence of O_2_ at ambient room temperature [244]. The O_2_ and HCl vapors act as etching agents and epitaxial recrystallization of Ag atoms occurs at the NW-NW junction under the action of these etching agents. The transition of Ag atoms from one side to other leads a significant decrease in sheet resistance by merging the two nanowires together. The merged planes of the two nanowires can be seen in HRTEM image in Figure 22d–f. Jin and coworkers also demonstrated a method to improve the conductivity of the TCE by using chloride ions, which can efficiently remove the capping agent. They fabricated a composite TCE by coating PET with alginate followed by AgNW solution [209]. This method utilized CaCl_2_ as an enhancing agent where the Cl^−^ ions removed the PVP coating and decreased the contact resistance between nanowires. Furthermore, Ca^2+^ being a bivalent cation metal reacted with alginate to enhance the bonding between Alginate and AgNW and form a super hydrophobic gel [245,246]. The treatment reduced the sheet resistance from 300 Ω/sq to 50.3 Ω/sq at a transmittance of 94% and the fabricated composite electrode was highly stable with no observable change in sheet resistance after a tape test. Xiong et al. fabricated an extremely air-stable and conductive FTCE with 8.4 Ω/sq sheet resistance and 86% transmittance on PET by vacuum filtration and transfer method which has been illustrated schematically in Figure 22g [247]. In order to diminish the junction resistance, the AgNW coated PET was primarily cured with electroless welding. The fabricated film was dipped into a solution of glucose and silver–ammonia for an optimized time. A subsequent addition of conductive iongel film obtained by copolymerization of 1-vinyl-3-ethylimidazolium dicynamide and N,N-methylene-bisacrylamide in the presence of ammonium persulfate (APS) was used as a protective layer to enhance the stability of the FTCE in the air without any noticeable increase in sheet resistance as shown in Figure 22h. Recently, Hwang and group fabricated AgNW based TCE with outstanding thermal and ambient stability by encapsulating it with an ultrathin Al_2_O_3_ film (around 5.3 nm) via low-temperature (100 °C) atomic layer deposition [248]. The Al_2_O_3_-encapsulated TCE were stable even after annealing at 380 °C for 100 min and maintained their electrical and optical properties. The Al_2_O_3_ encapsulation layer also effectively blocked the permeation of water molecules, thereby enhancing the ambient stability when exposed to an atmosphere with a relative humidity of 85% at 85 °C for over 1080 h as shown in Figure 22i. The fabricated electrode had 90 ± 3% transmittance and 50 ± 3 Ω/sq sheet resistance with excellent stability against humidity as well as temperature under harsh conditions. The electrode was stable at 400 °C, while the electrode without Al_2_O_3_ degraded at 200 °C.

#### 3.2.7. Light Induced Welding

Light induced welding has been utilized as a simple and time efficient post-treatment method to improve AgNW based TCE performance. The intensity of the light and treatment time are two important parameters while using light induced welding methods. Some researchers have studied the effect of sunlight to enhance the properties of the TCE [249]. The light intensity was controlled by a solar simulator, and could be further enhanced by using light converging lenses. They exposed the fabricated electrode to sunlight for 4 h to enhance the NW-NW junction contact and the results were satisfactory (T ~87% and R_s_ < 20 Ω/sq). However these results were superior to the results of thermal annealing at 200 °C as presented in Figure 23a. Silver nanostructures are susceptible to light; various morphological changes have been observed when they are exposed to various light sources [250]. Lee et al. demonstrated that a selective laser nano-welding was much more efficient in terms of processing speed as well as thermal controllability, as compared to broadband lamp welding. Since lasers are monochromatic sources of light whereby photon energy can be controlled, the treatment temperature can be accurately controlled depending on the substrate used. The occurrence of surface Plasmon resonance of nanowires and enhancement in electrical properties at the NW-NW junction can be seen in the FE-SEM images in Figure 23b. It is worth mentioning that the results obtained from the laser nanowelding technique were far better than those resulting from annealing at 250 °C, as shown in Figure 23c. High intensity pulsed light sintering technique can also be used in order to attain good electrical conductivity in a short time period. Jiu et al. fabricated an AgNW based TCE on PET using Mayer rod coating followed by HIPL treatment to enhance the properties of the TCE by varying the light energy from 0.21–2.33 J/cm^2^ [248]. Improved contact at the NW-NW and NW-substrate junction was observed at 1.14 J/cm^2^, while the nanowire morphology was critically damaged at 2.33 J/cm^2^, as shown in Figure 23d. Light induced nanowelding is an effective treatment method for Ag nanowires to interconnect at the junctions. The light induced heat energy efficiently enhances the junction geometry in such a way that the planes of nanowires completely merge with each other due to the heat generation in highly compact AgNW network. A complete epitaxial recrystallization of Ag atoms to make twinned planes by merging two planes under plasmonic nanowelding at the NW-NW junction can be seen in the TEM and SAED patterns in Figure 23e–h [251]. The light exposure time span is a critical parameter when dealing with high energy irradiation. In order to minimize the thermal damage, an ultrafast femtosecond laser nanowelding of AgNWs was demonstrated [252]. Limited thermal diffusion and localized field enhancement combined with femtosecond lasers enables the maintenance of the original crystal structure while enhancing junction electrical properties. The fabricated electrode after enhancement results in 94% transmittance and 25 Ω/sq sheet resistance. A comparative study with other heating methods is shown in Figure 23j. Recently, research efforts have also been made to accomplish fast AgNW sintering at low cost using a xenon flash lamp [158]. Improved optoelectronic properties on a stretchable PDMS electrode were achieved after xenon flash lamp treatment as shown below in Figure 23i.

## 4. Conclusions and Future Outlooks

Herein, we discussed the various AgNW synthesis protocols and strategies used for fabricating TCE with improved optoelectronic properties. In recent years, the polyol method has been widely used for AgNW synthesis with good control over nanowire morphology. While this method possesses various advantages like simplicity and scalability, the use of organic surfactants like PVP can degrade the optoelectronic performance of the fabricated TCE, resulting in a hydrophilic surface. Furthermore, optimization of the polyol process is critical to ensure that all Ag seeds are transformed into nanowires, as these seeds can agglomerate together with the nanowires after centrifugation. Besides centrifugation, researchers have also used cross flow purification and decantation to remove organic and inorganic contaminants [255,256]. Some researchers have utilized millifluidic reactor systems for AgNW synthesis in order to better control the chemical potential, supersaturation state and nanowire aspect ratio [127,257]. Characterization techniques like X-ray adsorption spectroscopy (XAS) have also been utilized to investigate the growth mechanism and reaction dynamics during AgNW synthesis, thus enabling improved control over the final nanowire morphology [258]. Furthermore, efforts have been made to enable green AgNW synthesis where the chemical reagents are biocompatible and can be easily disposed of [259]. Since AgNWs are prone to oxidation in air, researchers have also tried coating them with polymers and anti-oxidative metals like gold by making core shell structures [260,261,262,263,264,265]. While AgNW based TCE have several advantages including scalable solution based processing capability, low sheet resistance, high conductivity, low surface roughness and mechanical flexibility and stretchability, there are still some issues that need to be resolved; for example, the non-uniform distribution of the AgNW network obtained during solution based deposition can lead to localized joule heating, which can degrade optoelectronic performance [266]. Optimization of deposition parameters is key to ensure uniform AgNW distribution on the substrate surface. Finally, progress and optoelectronic enhancement of AgNW based TCE will enable a range of applications, from flexible optoelectronics to wearable devices [267,268]. While this review is primarily focused on synthesis of AgNW and techniques to fabricate AgNW based TCE, there has also been widespread research interest in utilizing these TCE for optoelectronic applications, including electronic skins [19,118], smart clothes [20], organic field effect transistors [269,270], wearable heaters [21,22], organic memory devices [271,272], flexible display panels [28], foldable displays [23,24,25,26], organic solar cells [219,273], smart windows [17,18], organics light emitting diodes [274,275], light emitting electro chemical cells [26,276] and compliant solar cells [27], among others. In summary, owing to their superior thermal and electrical conductivity, AgNW based TCEs are expected to play a key role in the future in a range of novel nano-optoelectronic applications.

## Figures and Tables

**Figure 1 nanomaterials-11-00693-f001:**
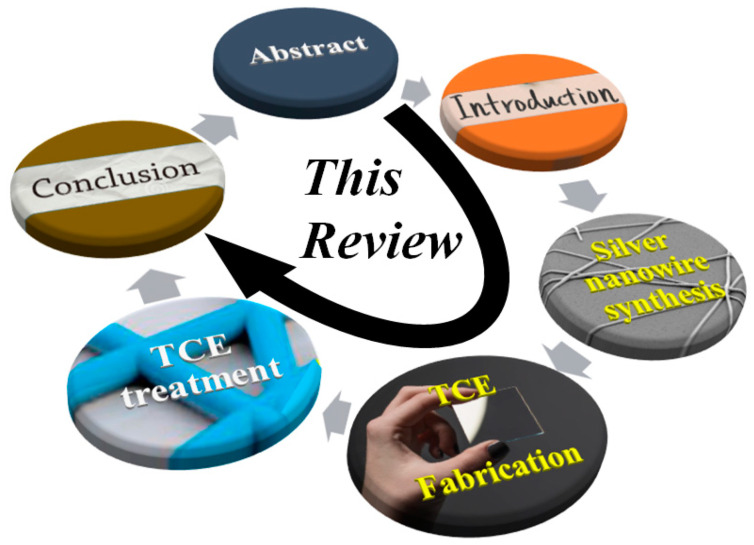
Schematic illustration of the review outline.

**Figure 2 nanomaterials-11-00693-f002:**
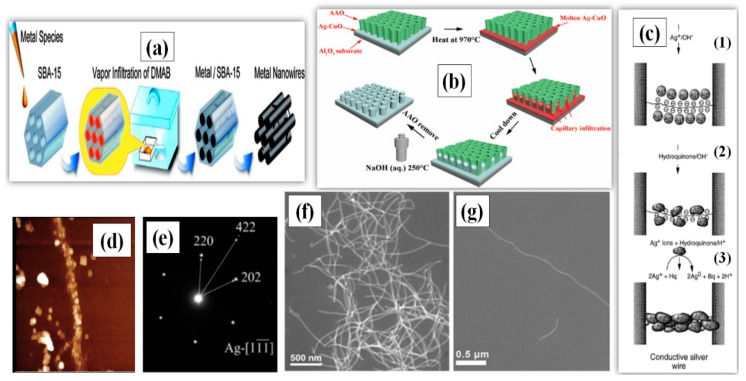
(**a**) A Schematic procedure for AgNW synthesis using Mesoporous Silica (SBA-15) through Vapor Infiltration Method (adapted from [43] with permission from ACS publications, 2010.) (**b**) A procedure for AgNW arrays growth within AAO template (adapted from [54] with permission from Elsevier, 2020). (**c**) A Schematic for AgNW synthesis using *λ*-DNA (C1), Silver-ion-loading on DNA Bridge. (C2), Metallic silver aggregates bound to the DNA skeleton. (C3), fully developed silver wire after HQ reduction. (**d**) Atomic force microscopy (AFM) images (Dimension 3000, Digital Instruments) of a silver wire connecting two gold electrodes 12 µm apart (adapted from [49] with permission from Springer nature, 1998). (**e**) Selected area electron diffraction (SAED) of AgNWs synthesized using AAO template (adapted from [54] with permission from Elsevier, 2020). (**f**) SEM images of AgNWs and (**g**) Scanning electron microscopy (SEM) image of a single AgNW synthesized using SBA-15 template adapted from [44] with permission from Springer nature, 2012).

**Figure 3 nanomaterials-11-00693-f003:**
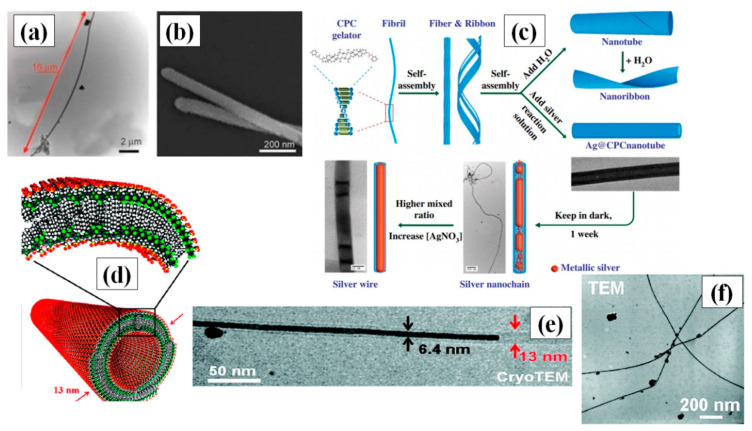
(**a**) TEM images confirming the maximum width and length of the AgNWs (**b**) SEM image of synthesized AgNW (adapted from [60] with permission from Express Polymer Letters, 2018). (**c**) Schematic diagram of the formation of CPC nanotubes and their aid in AgNW growing via a classical mirror reaction (adapted from [59] with permission from John wiley and sons, 2004). (**d**) Illustrating the self-assembled double walled nanotubular J-aggregates dyes with an outer diameter of 13 nm (**e**) Transmission electron microscopy (TEM) image of a AgNW with a width of 6.4 nm partially filling a supramolecular dye nanotube, and a silver nanoparticle on the nanotube’s outside 15 min after adding AgNO3 to the solution, (**f**) TEM image of synthesized AgNWs using J-aggregate, (72 h after adding AgNO_3_) (adapted from [62] with permission from ACS publications, 2010).

**Figure 4 nanomaterials-11-00693-f004:**
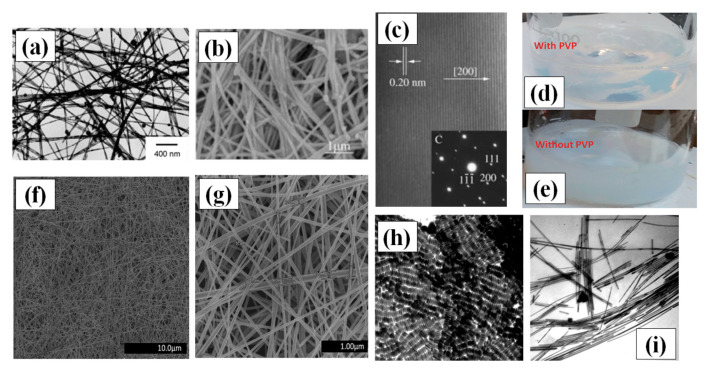
(**a**) TEM images synthesized AgNWs using 1,3-bis(cetyldimethylammonium) propane dibromide (adapted from [73] with permission from Elsevier, 2006) (**b**) SEM image of the prepared AgNWs using glucose as reducing agents (adapted from [72] with permission from Chemistry-A European Journal, 2005). (**c**) A HRTEM image and single crystal SAED pattern of AgNWs (**d**–**e**) Digital camera images of solutions prepared for hydrothermal synthesis (**d**) with PVP in the form of hydrosol and (**e**) without PVP having precipitates in solution (**f**,**g**) The SEM images of uniform, ultralong and thin AgNWs prepared by the hydrothermal method at 160 °C for 22 h of reaction (adapted from [74] with permission from Royal Society of Chemistry, 2013) (**h**) TEM image of shape controlled silver nanorods (using pH~11); scale bar = 100 nm. (**i**) TEM image of AgNWs (using pH~11); scale bar = 100 nm (adapted from [65] with permission from Royal Society of Chemistry, 1996).

**Figure 5 nanomaterials-11-00693-f005:**
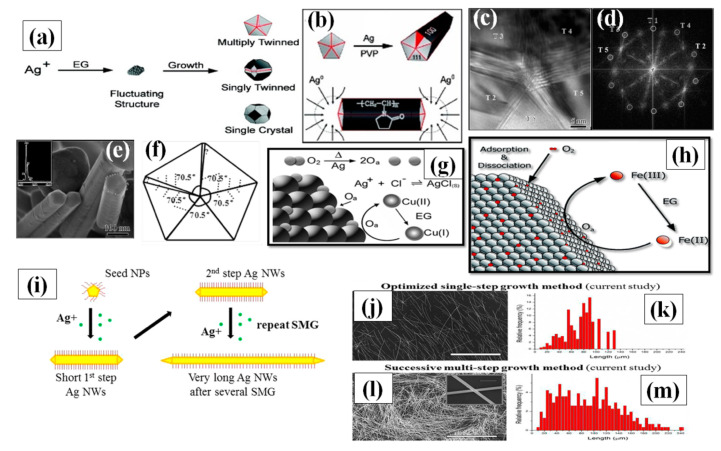
(**a**) the process of Ag^+^ reduction and various types of seeds (multiply twinned, singly twinned, or single-crystal) formation in polyol method. Therefore, one must regulate the crystallinity of the seeds in a reaction to produce a specific shape. (**b**) Once twinned decahedral seeds lengthened into rods. PVP selectively adsorbed on the {100} side facets so that Ag atoms could only add to the {111} facets at the ends of each rod (adapted from [90] with permission from ACS publications, 2007). (**c**) HRTEM image of the center area of an Ag cross-section showing five distinctive {111} twinning boundaries, (scale bar 5 nm). (**d**) FFT image of Figure 5c (**e**) pentagonal profiles of the nanorods formed in initial stage of synthesis. The inset shows the EDX data from a nanorod. (**f**) Pentatetrahedral twin model for an Ag nanorod formation in initial stage (adapted from [97] with permission from ACS publications, 2004). (**g**) A schematic illustration depicting the role of Cu-containing salts in the polyol synthesis of AgNWs has been shown. Oxygen present during initial seed formation can adsorb and dissociate on the Ag seeds, blocking sites for further Ag deposition. Cu (I) rapidly scavenges this adsorbed atomic oxygen (O_2_), with the Cu(I) being oxidized to Cu(II). Ethylene glycol (EG) reduces the Cu(II) back to Cu(I), providing a constant means for oxygen removal (adapted from [88] with permission from Royal Society of Chemistry, 2008) (**h**) The same mechanism Illustration of Fe-salt to removes atomic oxygen ref. (adapted from [89] with permission from ACS publications, 2005) (**i**) illustration of polyol process and a successive multistep growth process (**j**,**k**)) long AgNWs by an optimized single step growth conditions developed (**l**,**m**) very long AgNWs by the successive multistep growth (SMG) optimized process. All scale bars in SEM images are 100 μm and a scale bar in inset of panel L is 4 μm (adapted from [102] with permission from ACS publications, 2012).

**Figure 6 nanomaterials-11-00693-f006:**
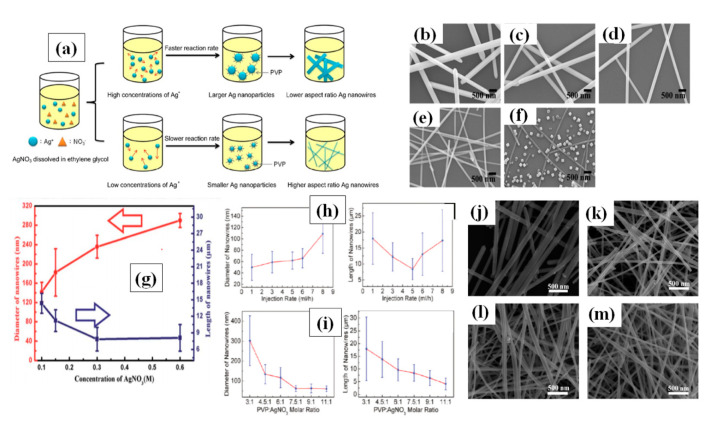
(**a**) A schematic of the different concentrations of silver nitrate to change the AgNW diameter. The synthesis of AgNWs using different concentrations of silver nitrate; (**b**–**f**) PVP/silver nitrate molar ratios are (**b**) 2, (**c**) 4, (**d**) 8, (**e**) 16, and (**f**) 32, respectively. (adapted from [103] with permission from Elsevier, 2015) (**g**) Changes in nanowire diameter and length as a function of the AgNO_3_ concentration (adapted from [104] with permission from Royal Society of Chemistry, 2014). (**h**) Shows the Changes in nanowire diameter and length with different injection rate. (**i**) shows the changes in nanowire diameter and length with PVP:AgNO_3_ molar ratio. (adapted from [101] with permission from ACS publications, 2012) (**j**–**m**) FE-SEM images of AgNWs synthesized with PVP molecules of different chain lengths (**j**) PVP-55,000, (**k**) PVP-360,000, (**l**) PVP-1,300,000, and (**m**) PVP-1,300,000 + PVP-55,000 (a weight ratio of 1:1) (adapted from [105] with permission from ACS publications, 2012))

**Figure 7 nanomaterials-11-00693-f007:**
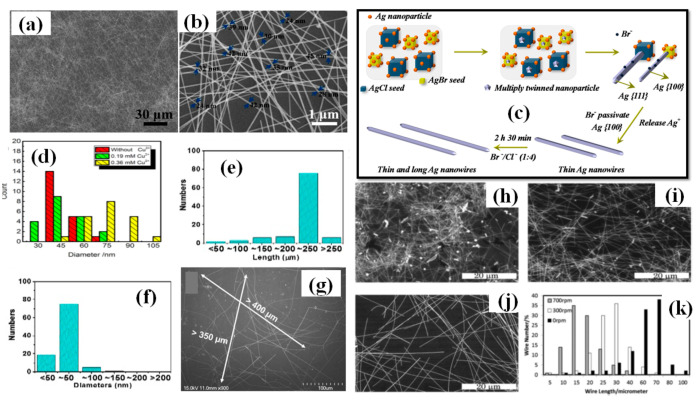
(**a**) Low-magnification FE-SEM image of the synthesized AgNWs; and (**b**) representative high-magnification FE-SEM image of the synthesized AgNWs (**c**) A Schematic representation of the proposed formation mechanism of high aspect ratio AgNWs using Br-ions (adapted from [113] with permission from MDPI, 2019) (**d**) A statistical analysis diameter of AgNWs under different condition (adapted from [114] with permission from Elsevier, 2016).The distribution of (**e**) length, and (**f**) diameters in a count of 200 Ag NWs (**g**) SEM images of synthesized AgNWs (scale bar 100 µm) (adapted from [111] with permission from ACS publications, 2017). (**h**–**j**) The SEM images of AgNWs prepared at 130 °C with stirring speeds of (**h**) 700, (**i**) 300 and (**j**) 0 rpm (the inset photos have a scale bar of 30 μm. (**k**) Statistics of the length distribution of AgNWs with different stirring speeds (adapted from [117] with permission from Royal Society of Chemistry, 2013).

**Figure 8 nanomaterials-11-00693-f008:**
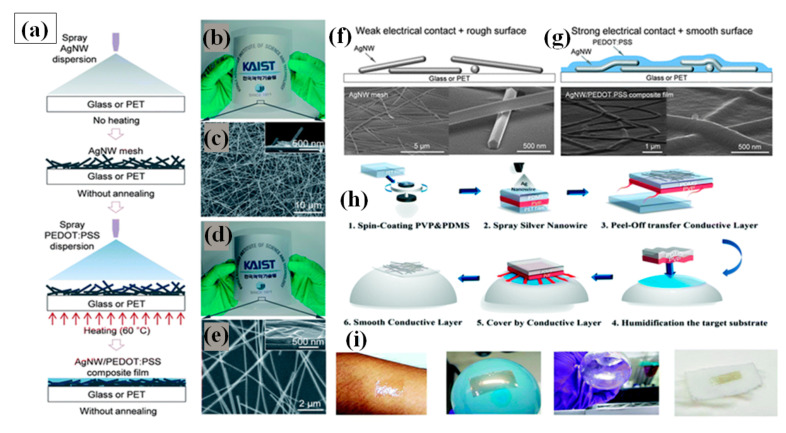
(**a**) Schematic diagram of the fabrication of AgNW–PEDOT:PSS composite electrodes through successive spray coatings. (**b**) AgNW mesh deposited on a flexible PET substrate by spray coating and its SEM image. (**c**) The inset image shows the cross-sectional view of the AgNW network. (**d**) AgNW–PEDOT:PSS composite film on a PET substrate with 81.7% (at 550 nm) of transmittance and 11.3 Ω/sq of sheet resistance. (**e**) SEM image of the AgNW–PEDOT:PSS composite film and a cross-sectional inset image. SEM image of AgNW film before (**f**) and after (**g**) PEDOT:PSS spray coating. (adapted from [144] with permission from Royal Society of Chemistry, 2013)Schematic of the Water Transfer technique process for AgNW TCEs. (**h**) (1) A prefabricated sacrificial layer (PVP) and a flexible framework (PDMS); (2) spraying of AgNWs on the top of PDMS after plasma treatment; (3) peel off the fabricated PVP-PDMS and AgNW film (4–5) transferred nanowire devices on a prehumidification target substrate with a sacrificial layer in between. (6) final film on random substrate (**i**) The corresponding photographs of various nonconventional substrates, including skin, a balloon, a round-bottomed flask, and an elastomeric textile (adapted from [149] with permission from Royal Society of Chemistry, 2019)

**Figure 9 nanomaterials-11-00693-f009:**
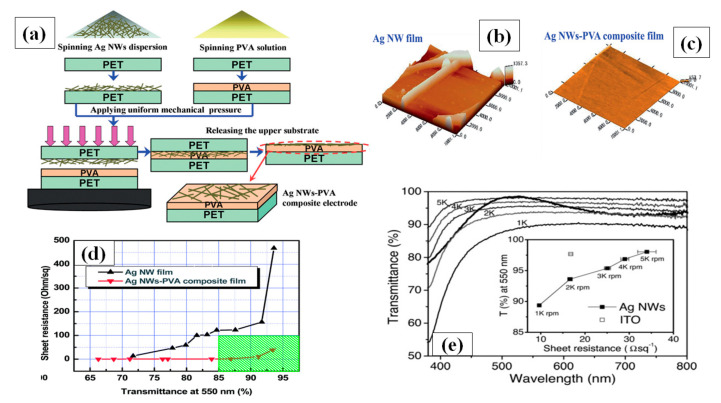
(**a**) A schematic procedure for AgNW/ PVA thin film fabrication by spin coating, (**b**–**c**) surface morphology of (**b**) bare AgNW and (**c**) AgNW/PVA composite film on PET (**d**) Transmission-resistance graph for AgNW and AgNW/PVA composite electrodes (adapted from [156] with permission from Royal Society of Chemistry, 2014) (**e**) a transmittance vs. resistance graph for different spin coating speed (1000–5000 rpm) (adapted from [153] with permission from John and Wiley sons, 2011).

**Figure 10 nanomaterials-11-00693-f010:**
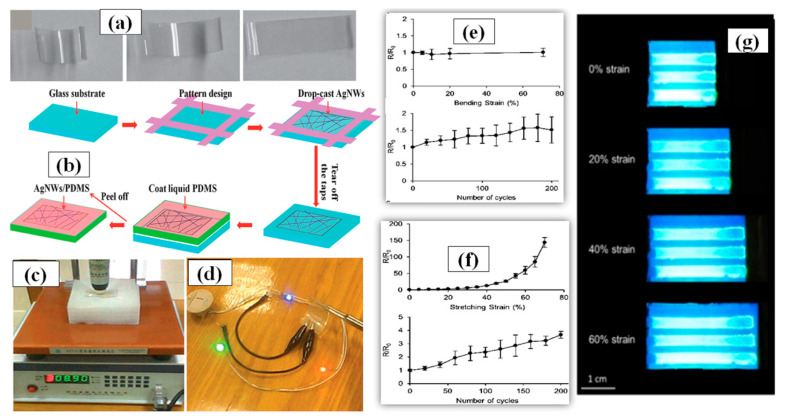
(**a**) illustrating the shape memory property of an AgNW/polymer electrode: photographs of a highly curved AgNW/polymer electrode (left) and its steady relaxation at 120 °C to an intermediate (center) and recovered shape after some time (right). (adapted from [160] with permission from John and Wiley sons, 2011) (**b**) A schematic procedure for AgNW/PDMS electrode fabrication by drop casting, (**c**) sheet resistance of the transparent electrodes (T = 81%) measured using a four-point probe instrument; (**d**) flexible transparent conductive film (T = 90%) in an electrical circuit powering three LEDs. (adapted from [161] with permission from Royal Society of Chemistry, 2014))∆R/R versus (**e**) bending strain and (**f**) stretching strain of fabricated electrode (**g**) real time device testing with no decay up to 60% stretching (adapted from [162] with permission from ACS publications, 2019).

**Figure 11 nanomaterials-11-00693-f011:**
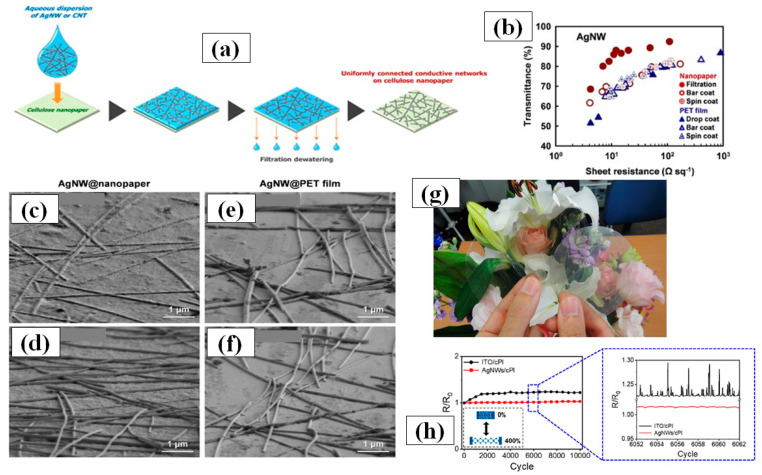
(**a**) Putative process of the formation of AgNW on nanopaper by drop and filtration coating, (**b**) A comparison between of optoelectronic properties of fabricated TCE between various methods. AgNW@nanopapers prepared by (**c**) filtration coatin and (**d**) bar coating and AgNW@PET films prepared by (**e**) drop coating and (**f**) bar coating (**g**) Optical images of original nanopaper (left), CNT@nanopaper (middle) and AgNW@nanopaper (right). Paper size: 75 mm in diameter. (adapted from [166] with permission from Springer Nature, 2014) (**h**) R/R_0_ vs no. of cycles with 0 to 400% strain each time of ITO/cPI and AgNW/cPI (adapted from [118] with permission from ACS publications, 2019).

**Figure 12 nanomaterials-11-00693-f012:**
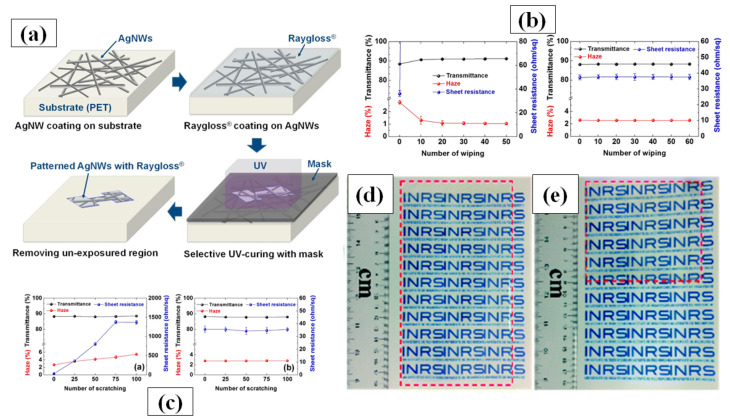
(**a**) Illustrations of the patterning process of Raygloss^®^/Ag nanowire electrodes. (**b**) Change in optical transmittance, haze, and sheet resistance as a function of the number of scratches for the bare Ag nanowire electrodes and Raygloss^®^/Ag nanowire electrodes, (**c**) The wiping test results of the bare Ag nanowire electrodes and Raygloss^®^/Ag nanowire electrodes (adapted from [174] with permission from Elsevier, 2017). Photographs of a large-area (18 cm × 9 cm) and flexible Ag NW TCE (**d**) before and (**e**) after bending by 180°. Red dashed lines highlight the edges of the flexible Ag NW TCE (adapted from [176] with permission from Royal Society of Chemistry, 2020).

**Figure 13 nanomaterials-11-00693-f013:**
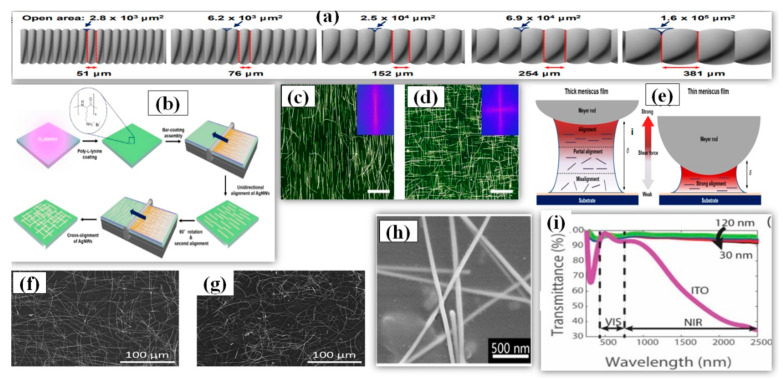
(**a**) Schematic illustration of the specifications of various Mayer rods. The rod number determines the diameter of wrapped wires and the open area. (**b**) A Schematic of procedure for the solution-processed bar-coating assembly to fabricate unidirectional and cross-aligned AgNW arrays. Before the bar-coating assembly, PLL solution is coated on O_2_ plasma-treated target substrate to form amine groups on the surface. Consecutively, the meniscus dragging produces highly aligned AgNW arrays. The cross-aligned AgNW array is formed by repeating the bar-coating assembly in a perpendicular direction to that of the pre-aligned AgNW arrays. Fast Fourier transform (FFT) analyses of the optical micrographs, indicating the direction and uniformity of the aligned AgNW structures. (**c**) Unidirectionally aligned and (**d**) cross aligned AgNW arrays. The scale bars are 40 μm. (**e**) Schematic showing the intensity distribution of shear force according to the variation of the height (*δ*) of meniscus (adapted from [180] with permission from ACS publications, 2017). SEM images of the AgNW networks deposited by (**f**) Mayer rod coating and (**g**) spray coating. R_sh_ is the sheet resistance while the stated transmittance (T) is at 550 nm and relative to the transmittance of the bare glass (adapted from [122] with permission from Royal Society of Chemistry, 2020). (**h**) SEM image of AgNW bound by PEDOT:PSS, the illustration shows the high uniformity and continuous of the fabricated composite FTCE (adapted from [183] with permission from Royal Society of Chemistry, 2019) (**i**) Transmittance vs. wavelength for AgNW networks at nanowire diameters of 30 nm, 70 nm, 100 nm, and 120 nm. The transmittance of AgNW networks are much greater in the near-infrared region than commonly used indium tin oxide (adapted from [184] with permission from Royal Society of Chemistry, 2020).

**Figure 14 nanomaterials-11-00693-f014:**
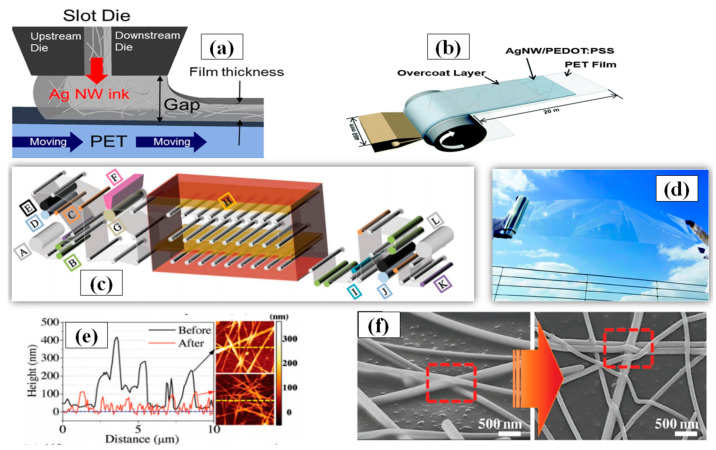
(**a**) schematic structure of a slot die head used for Ag NW coating with the influencing parameters (adapted from [188] with permission from John and Wiley sons, 2018) (**b**) A Schematic diagram of slot-die roll-coating process and structure of the AgNW/PEDOT:PSS conducting film (adapted from [187] with permission from Royal Society of Chemistry, 2015) (**c**) Roll to Roll slot-die coating of AgNWs: (**a**) schematic of R2R slot-die coating with calendaring process machine (A: unwinder, B: edge position controller, C: load cell, D: infeeder, E: nip roll, F: slot-die, G: backup roll, H: far-infrared dryer, I: accumulator, J: outfeeder, K: dancer, L: rewinder (adapted from [189] with permission from Springer nature, 2016) (**d**) Photographic image of the AgNW/PEDOT:PSS conducting roll film fabricated in this study (adapted from [187] with permission from Royal Society of Chemistry, 2015) (**e**) AFM data before and after calendaring (**f**) SEM image of AgNWs before and after calendering process. (adapted from [189] with permission from Springer nature, 2016).

**Figure 15 nanomaterials-11-00693-f015:**
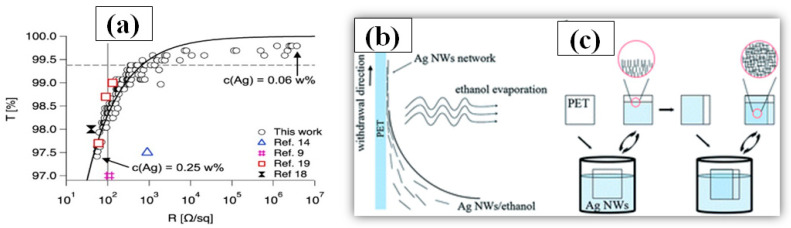
(**a**) The optical transmission T versus the sheet resistance R_s_. (adapted from [192] with permission from Springer nature, 2016) (**b**) Schematic illustration of nanowire assembly in the dip-coating process. (**c**) Two-step dip-coating process after rotating the substrate in perpendicular direction (adapted from [193] with permission from Royal Society of Chemistry, 2015)

**Figure 16 nanomaterials-11-00693-f016:**
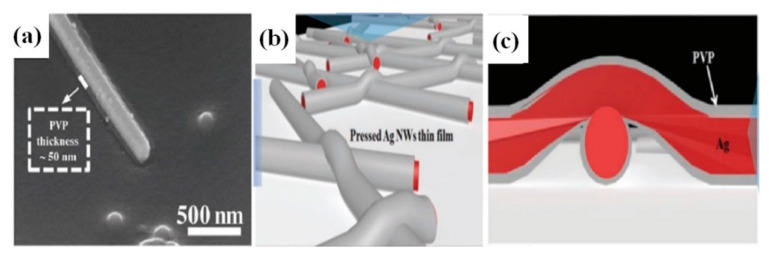
(**a**) SEM images of AgNWs surrounded by PVP layer of ~50 nm thickness (**b**,**c**) the concept of treatment for sheet resistance reduction (adapted from [188] with permission from John and Wiley sons, 2018).

**Figure 17 nanomaterials-11-00693-f017:**
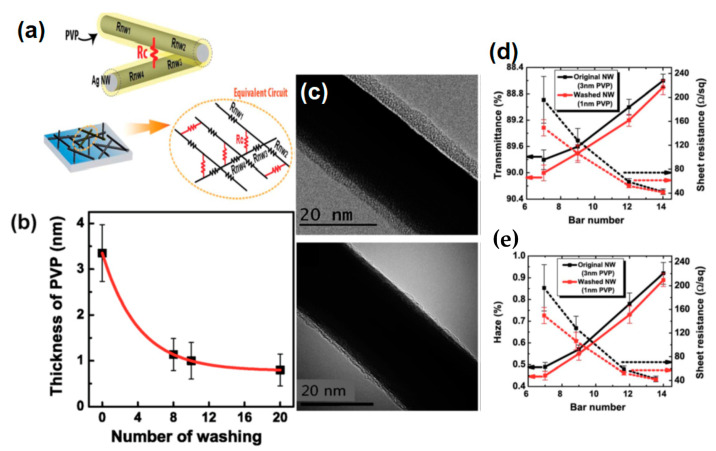
(**a**) Conceptual illustration of contact resistance and nanowire resistance, and the equivalent circuit of AgNW network which can be used for Monte Carlo simulation. (**b**) Average thickness measured for the PVP capping layer on the AgNWs with different numbers of repetitions of the washing process. (**c**)TEM image of an as-synthesized PVP-capped AgNW and the same NW after washing 20 times with DI water. (**d**) Transmittance versus bar number of a Mayer rod with error bars (**e**) Haze versus bar number of a Mayer rod with error bars (the values of R_S_ of the corresponding samples are plotted together in order to indicate the different areal coverage of the AgNW networks that were coated with different bar numbers of the Mayer rod). (adapted from [208] with permission from Royal Society of Chemistry, 2016)

**Figure 18 nanomaterials-11-00693-f018:**
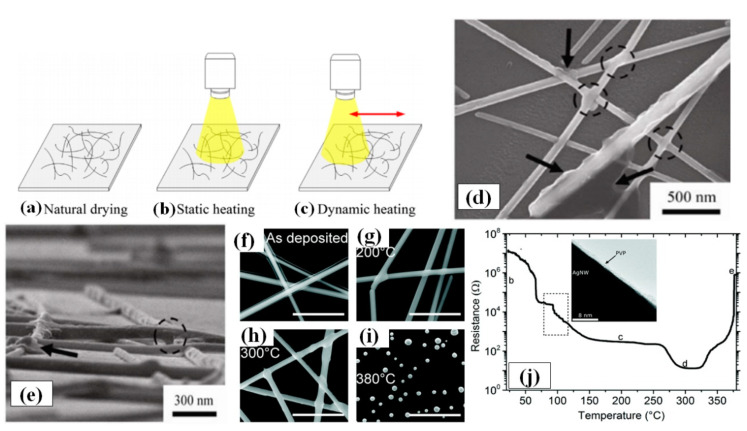
Schematic diagram of various drying methods. (**a**) Natural drying. (**b**) Static heating, (**c**) Dynamic heating (adapted from [213] with permission from Springer nature 2019). FE-SEM images of AgNW electrodes heated at 200 °C for 20 min having a sheet resistance of 9.5 Ω/sq. (**d**) Melting and fusion of AgNWs (inside the circles) and the remaining residues (marked by arrows) can be observed in the top view. (**e**) Tight connections of the AgNWs can be observed in the off-angle cross-sectional view (adapted from [214] with permission from Springer nature 2011). (**f**–**i**) Scanning electron microscope (SEM) images of: (**f**) as-deposited sample and of specimen annealed for 10 min at different temperatures: (**g**) 200 °C, first occurrence of observable sintering; (**h**) 300 °C all junctions are sintered; (**i**) 380 °C complete spheroidization of the network. Image (**h**) is at a lower magnification than the others to demonstrate that although the nanowires are completely spheroidized the resultant nanoparticles are still aligned at the original position of the wires. The scale bars in images (**f**,**g**,**i**) are 1 μm whereas that of image (**h**) is 4 μm. (**j**) Evolution of the electrical resistance of Ag nanowire network during a continuous thermal ramp of 15 °C min^−1^ from room temperature (associated with deposition solution of 0.75 mg mL^−1^ and an areal mass density of ≈105 mg m^−2^). The inset image is a TEM micrograph of a AgNW showing residual PVP from synthesis. (adapted from [215] with permission from Royal Society of Chemistry, 2014).

**Figure 19 nanomaterials-11-00693-f019:**
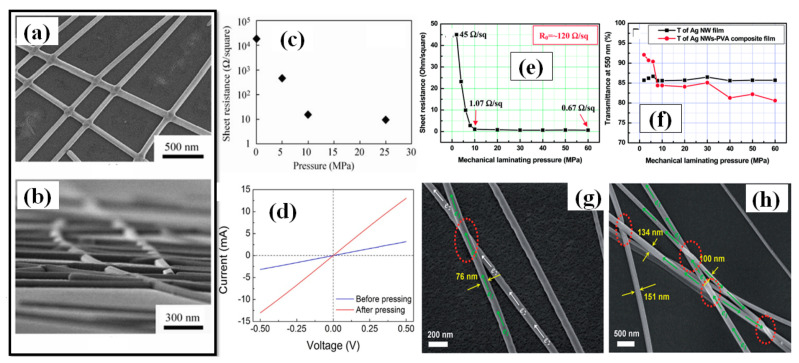
FE-SEM images of rinsed AgNW electrodes pressed at 25 MPa for 5 s (**a**) mechanically compressed AgNWs without any residues can be observed in the top view. (**b**) Tightly connected AgNWs with smooth surfaces can be observed in the off-angle cross-sectional view (**c**) Sheet resistance of a pressed AgNW electrode after pressing at various applied pressures (**d**) the I–V characteristics of Ag NWs before and after pressing. (adapted from [214] with permission from Springer nature 2011). (**e**) Relative plots of the sheet resistance versus the mechanical lamination pressure and (**f**) Relative plots of the transmittance at 550 nm versus the mechanical lamination pressure. (**g**) SEM images of the Ag NW film and (**h**) the Ag NW–PVA composite film on the PET substrate (adapted from [156] with permission from Royal Society of Chemistry, 2014).

**Figure 20 nanomaterials-11-00693-f020:**
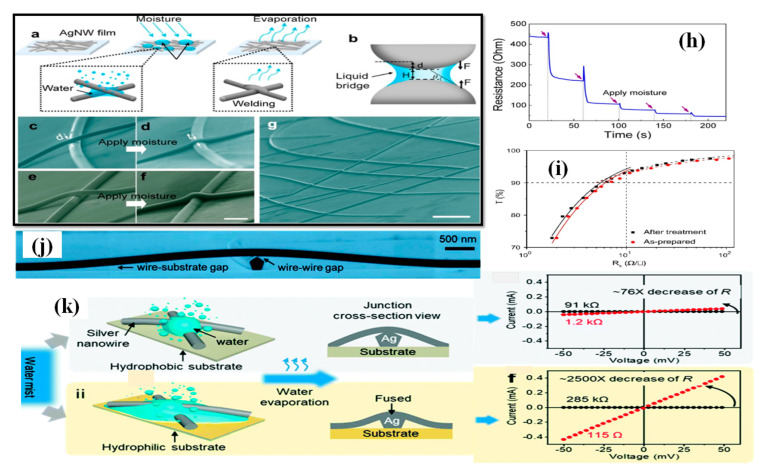
Capillary-force-induced cold welding of AgNWs. (**a**) Schematic of moisture treatment for capillary-force-induced cold welding of AgNWs. (**b**) Schematic of the mechanism of capillary interaction between two particles connected with a liquid bridge. (**c**–**f**) Two sets of SEM images of Ag wire-wire junctions before and after moisture treatment. Scale bar: 200 nm. (**g**) SEM image of a relatively large area of AgNWs showing well welded wire-wire junctions and relatively smooth surface caused by the moisture treatment. Scale bar: 1 μm (**h**) Resistance of a AgNW film significantly decreases after moisture treatment for a few cycles (adapted from [232] with permission from ACS publications, 2017). (**i**) A Plot of T versus log R_s_ for the AgNW-based TEs (adapted from [125] with permission from Elsevier, 2018). (**j**) SEM image of an AgNW wire–wire junction on a plain substrate. (**k**)) Illustration of the water-mist treatment process on different substrates for cold-welding) I-V curves of the AgNW junctions before (black dots) and after (red dots) water-mist treatment on hydrophilic glass (bottom) and hydrophobic PET (up) (adapted from [233] with permission from Royal Society of Chemistry, 2018).

**Figure 21 nanomaterials-11-00693-f021:**
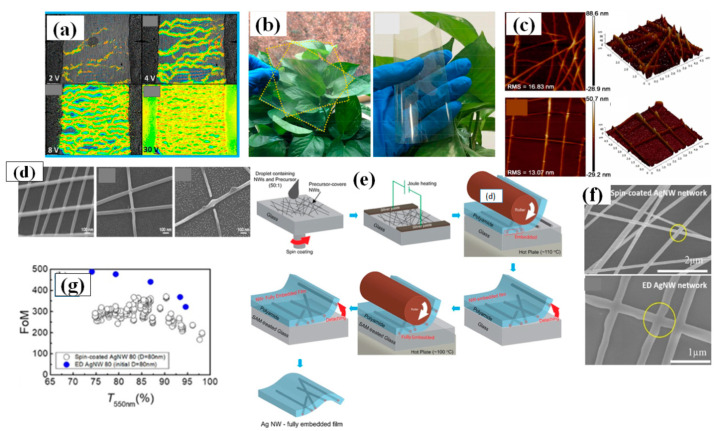
(**a**) Image shows the phase signal resulting from all the IR radiations collected during the 5 min cycles at a given voltage (2 V, 4 V, 8 V, 30 V) (adapted from [237] with permission from ACS publications, 2016). (**b**) Images of two stacked sheets of AgNW flexible electrodes and a sheet of AgNW flexible electrode which is folded through at 180°. (**c**) AFM images of a pristine AgNW network and regularly arranged AgNW network. The image size is 5 × 5 μm (**d**) SEM images of without treatment, with plasma treatment of 150 W for 2 min, 200 W for 5 min. (adapted from [238] with permission from Royal Society of Chemistry, 2020). (**e**) Process of transparent electrode fabrication from Ag NW mixing solution to Ag NW-fully embedded film (adapted from [239] with permission from ACS publications, 2019). (**f**) SEM images of the AgNW networks before and after electrodeposition. Electrodeposition welds the AgNWs together. (**g**) FoMs for AgNW networks with initial diameters 80 nm. The FoMs of the ED AgNW networks are superior to those of the as-spin-coated AgNW networks (adapted from [240] with permission from ACS publications, 2020).

**Figure 22 nanomaterials-11-00693-f022:**
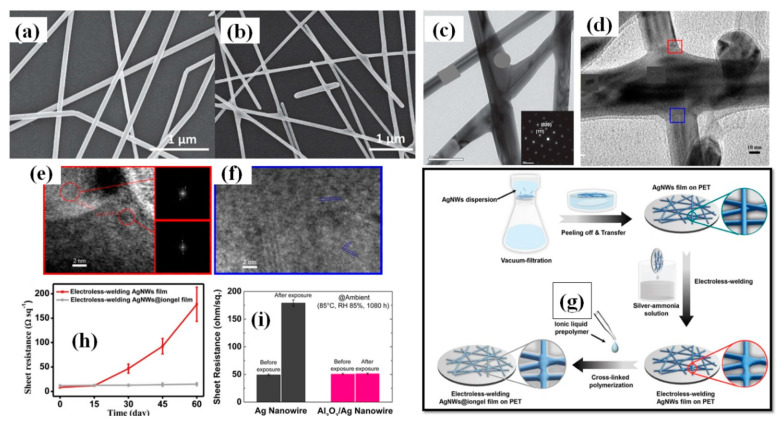
SEM images of transparent electrodes fabricated from 100 nm AgNWs (**a**) before and (**b**) after chemical treatment, (**c**) TEM image of crossed AgNWsafter chemical treatment better junction can be seen (adapted from [243] with permission from John and Wiley sons 2015). (**d**) Plane view TEM image of the welded junction between AgNWs. The frames represent the approximate positions of the welded junction for the lattice-resolved TEM in (**e**,**f**). Lattice-resolved TEM images of the upper (**e**) and lower (**f**) interfaces between the bottom AgNW and the junction, respectively. The upper right and lower right insets in (**e**) present FFT patterns from the bottom AgNW and the junction, respectively, which are different from each other. Lattice orientation of the junction in (**f**) mismatches that of the bottom AgNW. (adapted from [244] with permission from ACS publications, 2017). (**g**). Fabrication of stable AgNWs@iongel composite films for flexible transparent electrodes. (**h**) The sheet resistance of the electroless-welding AgNWs@iongel film shows tiny change even after exposure in air under ambient conditions for two months, while the sheet resistance of AgNW film without iongel sharply increases after 30 days(adapted from [247] with permission from John and Wiley sons 2016). (**i**) Sheet resistances of the Ag and Al2O3/Ag nanowire electrodes before and after exposure to ambient conditions with a relative humidity of 85% at 85 °C for 1080 h (adapted from [248] with permission from Springer, 2017).

**Figure 23 nanomaterials-11-00693-f023:**
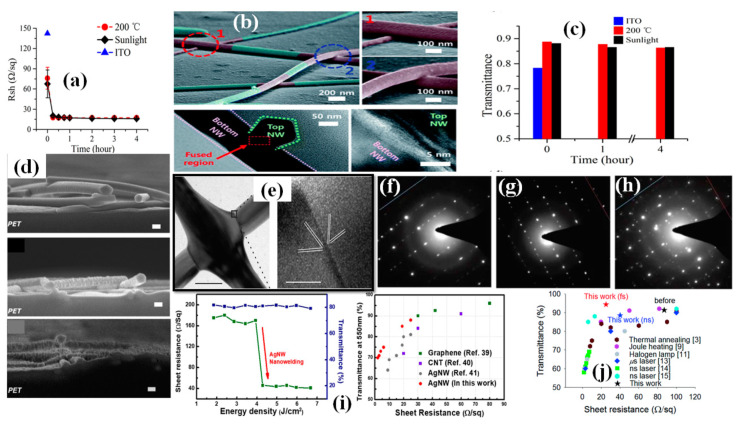
(**a**) Sheet resistance (Rsh) as a function of treatment time for samples treated with Sun sunlight illumination (black diamonds) and thermal annealing at 200 °C (red circles), respectively. (adapted from [249] with permission from Springer nature, 2017). (**b**) Magnified SEM (pseudo-colored) and HRTEM images Laser nano-welding of very long Ag NW network electrodes for highly transparent and flexible electrodes. (adapted from [164] with permission from Royal society of chemistry 2012). (**c**) Histogram of the light transmittance at 550 nm before and after different periods of time for both treatment methods (ref. [250])Cross-sectional SEM images of AgNW films on PET substrates (**d**) before and after HIPL sintering with light intensities of 1.14 and 2.33 Jcm^−2^, respectively. (adapted from [248] with permission from Springer nature, 2017). (**e**)TEM images of AgNW junctions after optical welding, merged planes can be seen in HRTEM image (**f**) SAED pattern of the top nanowire away from the junction. The large number of double diffractions caused by the pentagonally twinned cross-section lead to prominent lines of spots in only one direction, parallel to the blue line (**g**) SAED pattern of the bottom nanowire away from the junction. The double diffraction spots occur in a direction roughly perpendicular to that in b and parallel to the red line. (**h**) SAED pattern of the junction. Both sets of spots from b and c are visible. The double diffraction spots create a grid instead of an array of parallel lines as in g and h. (adapted from [251] with permission from Springer nature, 2012). (**i**) Sheet resistance and transmittance response to the flash light method induced nanowelding and Resulting sheet resistance and transmittance values compared to reference values from literature Improved optoelectronic properties on a stretchable PDMS electrode were achieved after xenon flash lamp treatment as shown in Figure 23i. (adapted from [253] with permission from Royal society of chemistry, 2016). (**j**) Comparison of the transmittance and sheet resistance with the previously reported results (adapted from [158] with permission from Springer nature 2018).

**Table 1 nanomaterials-11-00693-t001:** Summarized table of AgNW synthesis methods.

Method	Template	NW Length (µm)	NW Diameter (nm)	Ag Reduction Parameters	Characterization	Temperature and Time	Key Findings	Ref.
Hard template	* λ * -DNA	(12–16)	N/A	HQ/NaCl	FI (fluorescence imaging), AFM	N/A	a stretchable electrode in nanowire form	[49]
	SBA-15	>1	7	DMAB	TEM, EDS, XRD	40 °C for four days	nanowire length can be controlled by red. time	[43]
	SBA-15	4	7	Modified SBA-15	TEM, XRD, SEM	85 °C overnight	dodecanthiol as surfactant to avoid agglomeration	[44]
	AAO template	>20	50	High temp. melting to get Ag ions	XRD, SEM, EDS, BSE, TEM, DSC	970 °C for 30 min	novel hydrothermal purification step	[54]
Soft template	CHQ nanotubes	~1	~1	UV radiation, CHQ nanotube	TEM, EDS	NTP	ultrathin nanowires with long term stability	[55]
	*β*-CD	<20	~65	*β*-CD, FeCl_3_	TEM, EDS, XRD, SEM	170 °C for 1 h	FeCl_3_ addition facilitates > 300 aspect ratio	[60]
	J-aggregates of cyanine dyes	>10	7	nanotubular J-aggregate	UV-VIS, TEM	NTP	light enhanced growth of AgNW in dyes	[62]
	DHBC (PEO-b-PMAA)	>1	20–40	PMMA reducing agent	UV-VIS, TEM	NTP for 54 h	double polymers help in reduction and solubility	[64]
**Method**	**Surfactant/** **Red. Agent**	**NW Length** **(µm)**	**NW Diameter (nm)**	**Salt Mediator**	**Temperature and Time**	**Note**	**Ref.**	
Hydrothermal	PVP (40 k)/D+ glucose	200–500	45–65	NaCl	160 °C for 22 h	PVP avoids precipitation	[74]	
	Gemini surfactant	(10–90)	∼30	No salts used	100 °C for 24 h	Simple procedure at low temp. without using salts	[73]	
	glucose	<500	100	NaCl	180 °C for 18 h	Optimization of temp, time and reagent concentrations.	[72]	
Solvothermal	PVP (1300 k), EG	220	50	FeCl_3_	130 °C for 8 h	FeCl_3_ concentration optimization	[111]	
	PVP (1300 k), EG	120	40	KBr/NaCl (1:4)	170 °C for 2.5 h	KBr and NaCl ratio optimization	[113]	
	PVP(1300 k), glycerol	100–160	40–85	NaCl	150 °C for 5 h	AgNW synthesis without using transition metals and EG	[121]	
	PVP(1300 k), (propanediol)	140	140	diluted HCl	140 °C for 1 h	propendiol as a substitute of EG for rapid synthesis	[122]	
	PVP (360 k), EG	>100	100	CuCl_2_	130 °C for 3 h	AgNW and kirigami pattern combined for skin electronics	[118]	
	PVP(1300 k), EG	120	30	FeCl_3_, CuCl_2_, NaBr	170 °C for <1 h	Optimised Cu, Fe and Br conc. To attain 3000 aspect ratio	[123]	
Conventional polyol	PVP (360 k), EG	80–110	50–70	KBr, CuCl_2_	170 °C for 1 h	rapid synthesis with good control over dimensions	[124]	
	PVP (58 k), EG	123	120	FeCl_3_	130 °C for 3.5 h	high FoM of fabricated TCE	[125]	
	PVP(1300 k), EG	75	40	CuCl_2_	160 °C for 1.42 h	stirring and injection rate optimization	[126]	
Millifluidic polyol	PVP (55 k), EG	(10–20)	(50–100)	CuCl_2_	170 °C for 1.5 h	optimizing the millifluidic reactor parameter	[127]	

FI: fluorescence imaging; PVP: polyvinylpyrrolidone; DHBC: double hydrophilic block copolymer; *β*-CD: *β*-cyclodextrin; PMMA: poly (methyl methacrylate); C-HQ: calix-hydroquinone nanotubes; SBA: mesoporous silica; AAO: anodic aluminium oxide; NTP: normal temperature and pressure; EG: ethylene glycol; FoM: figure of merit; DMAB: dimethylamine borane;Gemini surfactant: 1,3-bis(cetyldimethylammonium) propane dibromide; Amphiphilic cyanine dye: 3,3-bis(2-sulfopropyl)-5,5,6,6-tetrachloro-1,1-dioctylbenzimidacarbocyanine (C_8_S_3_).

**Table 2 nanomaterials-11-00693-t002:** Summarized TCE fabrication and treatment method along with their comparison using FoM (Haacke’s formula).

Method	Substrate	Nanowire Dimensions	T (%) (*λ* = 550)	R_s_ (Ω/sq)	Fabricated Area (mm^2^)	FoM (10^−2^Ω^−1^)	Enhancement Process	Note	Ref.
Spray coating	Glass	L = 130 ± 36 μm, d = 124 ± 25 nm	91	5	N/A	7.7	annealing 200–300 °C	curved nanowires shape on glass due to spray coating	[122]
	Glass	L = 32–42 μm, d = 80 nm	90	15	812 × 812	2.2	solvent washing and annealing 230° C	PVP thickness reduced from 4 nm to 0.5 nm in five washing cycles	[210]
	PI	L = ~20 μm, d = ~50 nm	87.3	4.64	20 × 20	5.5	DC voltage, plasma treatment	very high flexible (500 cycle)	[238]
	PDMS	L = 25 μm, d = 30 nm	86	38	>100	0.6	photonic sintering using xenon lamp	Stable at 1mm bending radius, stretchable (50%)	[158]
Spin coating	Glass	L = 26–138 μm, d = 20–32 nm	89.2	2.9	25 × 25	31.9	annealing at 200 °C	high FoM (~0.32Ω^−1^)	[216]
	PET	L > 100 μm, d = 72 nm	86.9	0.75	30 × 30	32.4	mechanical pressing 30 MPa	AgNW-PVA film, R_s_ reduced from 120 to 0.75, haze 7.1%	[155]
	glass	L = 123 μm, d = 120 nm	90.3	6.4	N/A	5.6	cold welding in CTAB solution	significant reduction in R_s_ of TCE by dipping CTAB	[126]
	PI	L = 25 μm, d = 32 nm	92	10.9	1.92	3.98	joule heating	ultra smooth surface (1.92 nm), 19% R_s_ reduced	[239]
	glass	L = 100–150 μm d = 45–80 nm	87	5.4	25 × 25	4.6	electrodepositing using potentiostat	reduced surface roughness and R_s_ after electrodeposition	[240]
	PDMS	L = 10–50 μm, d = 100 nm	85	15	100 × 100	1.3	O_2_ and HCl nanowelding in light fluorescent	bending stability (4000 times), electromagnetic interference shielding	[244]
	PET	L = 10–20, d = 30–40 nm	94	25	20 × 20	2.1	femtosecond laser ablation	avoids substrate damaging and Spheroidisation	[252]
Drop coating	PET	L = 89.5 μm, d = 84.3 nm	90	15	N/A	2.2	solvent washing	no post treatment, pretreatment decrease PVP layer from 13.9 nm to 0.96 nm	[209]
	glass	L = 8 μm, d = 70 nm	80	8.6	3 × 3	1.2	annealing at 200 °C, mechanical pressing at 25 MPa	R_s_ reduced from 6.9 × 10^6^ to 9.5 Ω/sq	[214]
	PDMS	L = 90 μm, d = 60 nm	85	15	50 × 50	1.3	annealing at 200 °C	70% stretchable and 70% bending, fulfil all modern parameters	[162]
Vacuum filtration	glass	L = 100–500 μm d = 160 nm	89	9	N/A	3.5	low temperature laser nanowelding	T = (89–95)%, R_s_ = (9–65 Ω/sq)	[164]
	PET	L = N/A, d = 30–70 nm	86	8.4	N/A	2.6	AgNH_3_ Electroless-welding, cross-linked polymerization	No change in R_s_ after 60 days	[253]
	Nanofiber cellulose paper	*L* = 8 μm, d = 70 nm	88	12	25 × 25	2.3	hot pressing at 110 °C and (1.1 MPa)	Nanopaper act as a substrate and filter, avoid transfer step	[166]
Doctor blade coating	PET	*L* = 15 μm, d = 40 nm	90	22	N/A	1.6	CaCl_2_ and alginate treatment	Novel process, R_s_ change from 300 to 50 Ω/sq at T = 94%	[254]
	PET	*L* = 8 μm, d = 70 nm	83	19	50 × 50	0.817	high intensity pulsed light	optimisation of light intensity (1.14 Jcm^−2^) and time (50 µs)	[248]
	PET	N/A	88.3	35	N/A	0.83	Raygloss coating and UV-curing	raygloss coating enhance the Transmission (by 2%) and stability against scratches and wipes	[174]
Mayer rod	Glass	*L*=130 ± 36 μm, d = 124 ± 25 nm	94	5	N/A	10.8	annealing 200–300 °C	Rod-coated (T = 94%) TCE is more transparent than spray coated (T = 91) at same R_s_.	[122]
	PET	N/A	96	12	70 × 70	5.5	50 °C temperature maintained on PET substrate	AgNW-PEDOT:PSS composite stable against wiping bending, writing	[183]
	PET	*L* = 20 μm, d = 35 nm	95	21	200 × 200	2.9	cross aligned NW, no treatment	cross aligned fabrication using doctor blade was found superior than other fabrication methods	[180]
Roll to roll slot die coating	Glass	*L* =100–150 μm, d = 70–120 nm	90	9	>100	3.9	water mist cold welding	hydrophilic substrate enhance NW-substrate junction and reduce R_s_ (~76 times)	[232]
	PET	*L* = 20 μm, d = 57 nm	92, 96	~12, 21.9	460 × 20000	3.6	annealing 150 °C for 2	very vast AgNW-PEDOT:PSS roll able composite film of 9.2 m^2^ area	[187]
	PET	*L* = 18.7 μm, d = 22.6 nm	90	50	400mm wide rolling film	0.7	annealing at 120 °C and 80 °C, UV treatment	≤1% haze, roll to roll coating machine attached with annealing and UV-curable coating layer	[189]
	glass	L = 19 ± 9 μm, d = 25 ± 5 nm	97.5	55	75 × 25	1.4	annealing at 120 °C	ultra transparent electrode (97.5–99.8%), low haze ideal for transparent screen	[192]
	PET	L = 5–50 μm, d = 50 nm	92	35	N/A	1.2	mechanical pressing 20 MPa, annealing at 80 °C	two step dip coating with two different axis, to reduce R_s_	[193]
	glass	L = 7, 15 μm, d = 90, 60 nm	85	15	15.1	1.3	annealing between 180 and 240 °C	annealing (temp. > 240 °C) or (time > 4 h) leads to abrupt increase in R_s_	[194]

PET: polyethylene terephthalate; R_s_: Sheet resistance; T: transmittance; CTAB: Cetyl trimethylammonium bromide; PDMS: Polydimethylsiloxane; MPa: Mega Pascal; PI: polyamide; AgNH_3_: silver ammonia; PEDOT:PSS: Poly (3,4-ethylenedioxythiophene) polystyrene sulfonate.

## Data Availability

Data sharing not applicable.

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
