# Peer review of "Silver Nanowire Synthesis and Strategies for Fabricating Transparent Conducting Electrodes"

_nanomaterials, 2021, doi:10.3390/nano11030693_

Round 1
Reviewer 1 Report
This manuscript for a review entitled “Silver nanowire synthesis and strategies for fabricating transparent conducting electrodes” by Chuang et al. presents a comprehensive overview of the recent advances in the synthesis methods for Ag NWs and fabrication routes of transparent conductors using Ag NWs. The manuscript includes many useful information of the related topic. However, I recommend the following issues need to be addressed before the final publication. Many issues are related to formality rather than content of the manuscript.
1.Fig 1 can be improved for better visibility. Currently, the figure looks not so good.
2. In many figures, the aspect ratio of the many original figures has been distorted, which makes the figures ugly. They all need to be fixed not to change the original aspect ratio.
3. The authors should check the sequence of the sub-figures overall. The sequence should be from left to right, and from top to bottom, which is the very basic requirement.
4. Alignment of the subtitles in 2.2.2 should revised. There is too much space in front of the subtitles.
5. It is recommended that the authors need to add the marks (a, b, c….) all again for each figure with the same font design. Authors need to refer to other review papers to check how those things are edited without violating the copyright.
6. The authors should check the space between numbers and units. It is not consistent overall in the current version.
7. Each table should have its table number.
8. The nomenclatures at the bottom of the tables need to be aligned again.
9. Some part of the table at page 40 has been composed by a different font.
10. The authors need to check what is the difference between hyphen and dash. Then, hyphens and dashes should be checked overall in the manuscript.
11. Each equation should have its own equation number.
12. The authors need to check whether ‘Fig x.’ or ‘Fig x:’ in the caption of each figure.
13. In the table of contents, is there any reason why there is a large space in front of some parts of subsections?
Author Response
Dear Reviewer:
We have carefully replied all the comments in a separated file. please refer to the attached file.
Best Regards,
Cheng-Hsin Chuang Ph.D. Professor/Chair
Institute of Medical Science and Technology
National Sun Yat-sen University
70 Lienhai Rd., Kaohsiung 80424, Taiwan.
E-mail:chchuang@imst.nsysu.edu.tw Tel: +886-7-5252000 ext 5785
Mobile: 0937929830
Website:http://manstlab2015.blogspot.com/ ResearchGate: https://www.researchgate.net/profile/Cheng_Hsin_Chuang

Reviewer 2 Report
The manuscript gives a detailed description about synthesis and strategies for silver nanowire based transparent conductive electrodes. To make the manuscript more complete, several parts should be added to in the revised version.
1# In introduction, in addition to transparent conductive films, one-dimensional metal nanowires have been widely used in flexible and stretchable electronic devices for wearable electronics. Several recently published review papers should be cited to emphasize the importance of metal nanowires, such as https://doi.org/10.3390/nano8080628, https://doi.org/10.1021/acs.chemrev.8b00745, and https://doi.org/10.1002/admt.201800546
2# In 1.2.2, for obtain transparent conductive films with high optoelectronic property, silver nanowires with high aspect ratio are often required. As the main method to synthesize high aspect ratio silver nanowires, high aspect ratio silver nanowires (aspect ratio larger than 1000) synthesized by polyol method should be summarized in one table by emphasizing on the main improvement of synthesizing process, and the corresponding optoelectronic properties.
3# In order to coating methods, printing techniques have also been widely used to fabricate silver nanowire based (transparent conductive) electrodes or films. In 3.1, one chapter should be added. Some published papers about printing silver nanowires for the authors to refer. Inkjet printing (https://doi.org/10.1063/1.4913697,https://doi.org/10.1021/acsanm.8b00830) screen printing (https://doi.org/10.1002/adma.201600772) gravure printing (https://doi.org/10.1016/j.tsf.2015.04.055, https://doi.org/10.1038/s41598-018-33494-9) and capillary printing (https://doi.org/10.1021/acs.nanolett.5b03019)
4# Although this review focuses on synthesis and strategies for silver nanowire based transparent conductive electrodes, I still think several applications of such transparent conductive electrodes should be mentioned. Several paragraph about application should be added before Conclusion and Outlook.
Author Response
Dear Reviewer:
We have replied all the comments in a separated file, please refer the the attached file.
Best Regards,
Cheng-Hsin Chuang Ph.D. Professor/Chair
Institute of Medical Science and Technology
National Sun Yat-sen University
70 Lienhai Rd., Kaohsiung 80424, Taiwan.
E-mail:chchuang@imst.nsysu.edu.tw Tel: +886-7-5252000 ext 5785
Mobile: 0937929830
Website:http://manstlab2015.blogspot.com/ ResearchGate: https://www.researchgate.net/profile/Cheng_Hsin_Chuang

Reviewer 3 Report
The authors describe in their review entitled “Silver nanowire synthesis and strategies for fabricating transparent conducting electrodes” the major advantages of suitable optical transparent Ag nanowires for electrode applications. Although the review is basically based on synthetic approaches, it is a valuable addition to the review literature of transparent conducting electrodes. The manuscript itself is drafted in a logical way and can be further processed for publication.
Author Response
Dear Reviewer:
We have replied all the comments in a separated file, please refer to attached file.
Best Regards,
Cheng-Hsin Chuang Ph.D. Professor/Chair
Institute of Medical Science and Technology
National Sun Yat-sen University
70 Lienhai Rd., Kaohsiung 80424, Taiwan.
E-mail:chchuang@imst.nsysu.edu.tw Tel: +886-7-5252000 ext 5785
Mobile: 0937929830
Website:http://manstlab2015.blogspot.com/ ResearchGate: https://www.researchgate.net/profile/Cheng_Hsin_Chuang

Round 2
Reviewer 1 Report
The authors have properly addressed reviewers' previous concerns, comments and questions in the revised version. Therefore, I have no objection to the publication of this manuscript in Nanomaterials. The manuscript seems to have been improved a lot over the original submission.